# TGFβ blocks IFNα/β release and tumor rejection in spontaneous mammary tumors

Marion V. Guerin [1], Fabienne Regnier[1], Vincent Feuillet[1], Lene Vimeux[1], Julia M. Weiss[1,4], Georges Bismuth[1], Gregoire Altan-Bonnet [2], Thomas Guilbert[1], Maxime Thoreau[1], Veronica Finisguerra[3], Emmanuel Donnadieu[1], Alain Trautmann [1,5] & Nadège Bercovici[1,5]

Type I interferons (IFN) are being rediscovered as potent anti-tumoral agents. Activation of the STimulator of INterferon Genes (STING) by DMXAA (5,6-dimethylxanthenone-4-acetic acid) can induce strong production of IFNα/β and rejection of transplanted primary tumors. In the present study, we address whether targeting STING with DMXAA also leads to the regression of spontaneous MMTV-PyMT mammary tumors. We show that these tumors are refractory to DMXAA-induced regression. This is due to a blockade in the phosphorylation of IRF3 and the ensuing IFNα/β production. Mechanistically, we identify TGFβ, which is abundant in spontaneous tumors, as a key molecule limiting this IFN-induced tumor regression by DMXAA. Finally, blocking TGFβ restores the production of IFNα by activated MHCII[+] tumor-associated macrophages, and enables tumor regression induced by STING activation. On the basis of these findings, we propose that type I IFN-dependent cancer therapies could be greatly improved by combinations including the blockade of TGFβ.

[1] Université de Paris, Institut Cochin, INSERM, U1016, CNRS, UMR8104, F-75014, Paris, France. [2] Cancer & Inflammation Program, National Cancer Institute, Bethesda, MD, USA. [3] Ludwig Institute for Cancer Research, de Duve Institute, Université catholique de Louvain, Brussels, Belgium. [4] Present address: Division of Pediatric Hematology and Oncology, University Medical Center, Freiburg, Germany. [5] These authors contributed equally: Alain Trautmann, Nadège Bercovici. Correspondence and requests for materials should be addressed to N.B. (email: nadege.bercovici@inserm.fr)

  1

Type I interferon (IFN) α and β are cytokines with a great potential in anti-tumor immunity. It has been shown that endogenous type I IFN constitutes a first line of defense by innate cells, promoting an adaptive immune response not only against viruses but also against cancer cells[1–3]. In fact, modulation of immune responses by type I IFN may occur in several different ways. It often involves the regulation of cytokines and chemokines (CXCL10, CCL3, CCL2, and IL-15), which are able to promote the recruitment, survival, and activation of various immune cell subsets, including monocytes, dendritic cells (DC), T cells, and NK cells. Moreover, IFNα induces the upregulation of MHC I and the acceleration of DC differentiation, thus improving the priming of T cells[4,5]. In addition, the potential of IFNα/β to inhibit protein translation can trigger conflicting signals leading to the death of highly proliferating cancer or endothelial cells[6,7].

Long recognized as a broad immune modulator, type I IFN has been administered for years to treat cancer patients[8]. However, due to its toxicity after systemic administration, it is no longer commonly used as such. A better understanding of the mechanisms by which type I IFN modulates immune responses has renewed the interest for its use. Recently, the ER-associated molecule STING has been the focus of several investigations aiming at restimulating the production of type I IFN in the tumor ecosystem. STING is ubiquitously expressed and is activated by cytosolic nucleotides derived from pathogens or following cellular damages. In particular, it has been shown that STING can sense endogenous cytosolic DNA and contribute to anti-tumor immunity[9]. In particular, synthetic cyclic dinucleotides (CDN) injected intratumorally have been shown to rapidly elicit the expression of IFNβ by endothelial cells in tumors, followed by the generation of anti-tumor specific CD8+ T cells and the regression of transplanted tumors[10,11]. DMXAA (5,6-dimethylxanthenone-4-acetic acid) has the dual property of activating STING and disrupting specifically the tumor vasculature[12]. One intraperitoneal (i.p.) injection is sufficient to induce the regression of primary transplanted tumors, as a single agent or in combination with vaccination[13–15]. In line with this, we have recently reported that the DMXAA-induced regression of PyMT transplanted tumors relies on IFNα/β production and on the cooperation of T cells with myeloid cells at the tumor site[16].

Together, these reports suggest that inducing type I IFN in solid tumors by targeting the STING pathway is a promising therapeutic approach. Nevertheless, what is unknown is whether it would be sufficient to induce the rejection of solid tumors that arise spontaneously, since multiple resistance mechanisms take place during the progression of spontaneous tumors. In particular, the transplantation of tumor cells induces an acute inflammation locally associated with cell death and priming of the immune cell infiltrate[17]. In contrast, spontaneous tumors emerge from a sterile microenvironment progressively exposed to a chronic inflammation likely to condition differently the immune cell infiltrate. When spontaneous tumors develop in an animal, the T-cell repertoire shaped in the thymus and the periphery gives rise to low affinity and anergic tumor-infiltrating T cells[18]. In addition, soluble factors of chronic inflammation like TGFβ are known to suppress various immune effector cells, modulate the profile of myeloid cells, and promote the emergence of regulatory T cells. Thus, although transplanted tumors share some elements of immune suppressive environments, their rapid development, within days or weeks, do not integrate the various changes that occur in spontaneous tumors during months of development. In the MMTV-PyMT tumor model, Ming Li et al. have described the progressive accumulation of suppressive tumor-associated macrophages (TAM) subsets in the mammary tumors from 8-week-to 20-week-old transgenic mice[19]. The density of such TAM,

assimilated to "M2-like macrophages", which represent a major component of the tumor microenvironment, has been correlated with a poor prognosis in various cancer types in humans[20]. By contrast, detailed analyses of the myeloid cell compartment in human tumors revealed that the density of TAM with an M1-like phenotype correlates with a favorable prognosis in some cancers[21]. In line with this, when appropriately stimulated for instance in the presence of type I IFN, macrophages can mediate anti-tumor activity together with anti-tumor CD8+ T cells, at least in transplanted tumor models[16,22–26]. Thus, the nature and the activation status of the tumor microenvironment may drastically influence the outcome of therapeutic treatments.

In this study, we address whether targeting STING with DMXAA can induce the regression of spontaneous MMTV-PyMT mammary tumors. We identify TGFβ as a key element that impairs the production of IFN type I by TAM, preventing the induction of tumor regression.

## Results

**Limited efficacy of DMXAA in the spontaneous MMTV-PyMT mice.** To assess if tumors developing in MMTV-PyMT mice (Spont-PyMT) could regress after DMXAA treatment, as reported earlier for transplanted PyMT mice (Trans-PyMT)[16], Spont-PyMT mice were injected intraperitoneally (i.p.) with DMXAA when posterior tumors reached ~6 mm in diameter (in ~2-month-old mice). For each treated mouse, the size of 8–10 tumors was measured over time. Figure 1a provides an example of the follow-up obtained in one Spont-PyMT mouse treated with DMXAA. In the central panel, the raw data (one tumor per curve) illustrate the evolution of the tumor size for one mouse. In the right panel of Fig. 1a are shown the same data, but the size of each tumor is expressed relative to its size at day 0. This representation allows one to distinguish easily tumors that responded or not to DMXAA treatment. Figure 1b illustrates the global follow-up obtained in the groups of DMXAA- or DMSO-treated mice. Overall, DMXAA only slowed down tumor growth compared with control mice, and 5 days after the DMXAA injection, very few mice (14%) showed regressing tumors in Spont-PyMT mice, whereas 95% did in transplanted PyMT (Fig. 1c).

One explanation for this phenomenon could have been that DMXAA could not access Spont-PyMT tumors as it does in Trans-PyMT ones, due to putative differences in the vasculature of Spont-PyMT tumors and of Trans-PyMT ones. To bypass such a potential problem, we assessed if two different STING ligands, DMXAA or ML RR-S2 CDA, could induce the activation of the type I IFN pathway when the drug was injected directly into the tumor. The results obtained after i.t. injections confirmed those obtained after DMXAA i.p. Indeed, both i.t. STING agonists failed to induce tumor regression in Spont-PyMT mice, in sharp contrast with what was observed in the transplanted model (Fig. 1d).

To understand why DMXAA was poorly efficient in these Spont-PyMT mice, we examined the immune response triggered by DMXAA in Spont-PyMT tumors, compared with Trans-PyMT ones. The proportion of CD45+ immune cells infiltrating Spont-PyMT tumors was modestly increased after DMXAA injection (from 35% to 50% of viable cells, Fig. 2a; Supplementary Fig. 1). This was in sharp contrast with the massive infiltration measured in the Trans-PyMT tumors after DMXAA (from 22% to 86% of viable cells). By analyzing the immune cell subsets in more details, we observed a transient neutrophil (CD11b+ Ly6G+) infiltrate at 24 h after DMXAA injection (35–40% of the infiltrating cells) in both models. However, the population of TAM (CD11b+ Ly6C$^{neg}$ Ly6G$^{neg}$ F4/80+) persisted in spontaneous tumors after the injection of DMXAA, whereas it decreased

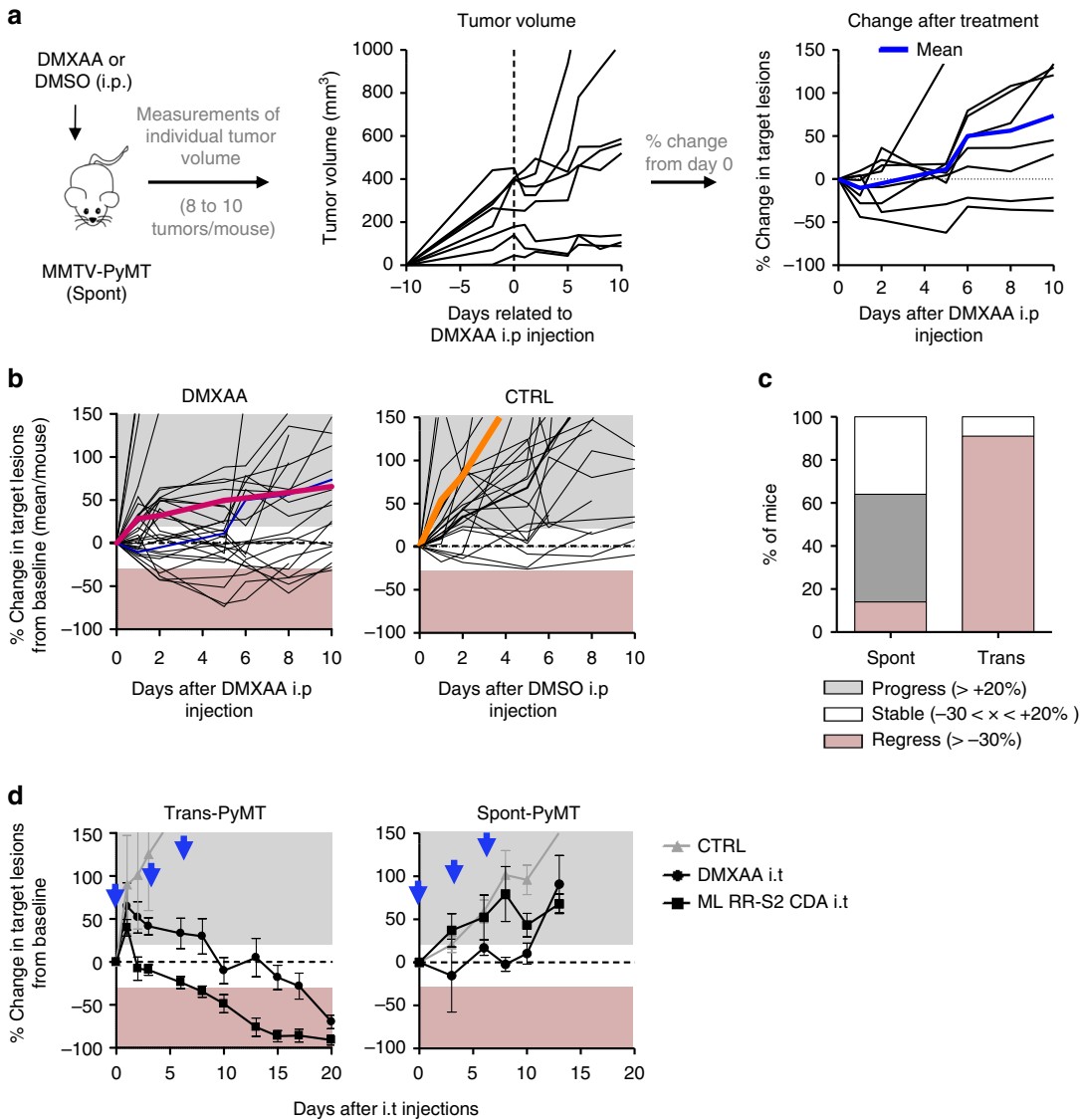

**Fig. 1** DMXAA does not induce a systematic regression of Spont-PyMT tumors. **a** Spont-PyMT mice were treated at day 0 with one i.p. injection of DMXAA or DMSO as a control. For each treated mouse, the changes in tumor size from baseline (d0) was calculated at each time point and expressed relative to the size at d0. The graphs in (**a**) provide an example of measurements obtained in one mouse treated with DMXAA i.p., with one curve per tumor. The average curve of all tumors of this mouse is shown in blue. The mouse clipart was created by N.B. **b** Each black curve corresponds to one mouse, and is calculated as the average relative tumor change for this mouse (as shown in **a**). The average curves for all the animals of a cohort are shown in red (DMXAA) or orange (CTRL). Overall progression (gray area) or regression (pink area) of tumor burden are indicated. Spont-PyMT mice: $n = 30$ (DMXAA), $n = 25$ (CTRL), from nine independent experiments. **c** The proportion of mice with, on average, regressing, stabilized or progressing tumors, are represented for these Spont-PyMT mice, in comparison with Trans-PyMT mice. Spont-PyMT mice: $n = 30$; Trans-PyMT mice: $n = 26$, from 9 to 10 independent experiments, respectively. **d** Intratumoral injection of DMXAA or ML RR-S2 CDA-induced tumor regression in Trans-PyMT mice, but not in Spont-PyMT mice. Trans- and Spont-PyMT mice were injected i.t. with DMXAA (1 × 500 µg, day 0) or ML RR-S2 (3 × 50 µg, at days 0, 3, and 6) or with HBSS i.t. as controls. Blue arrows: days of injections. The average fold change in tumor volume compared with baseline (day 0) is shown for each treated group. ML RR-S2 group (Spont-PyMT mice $n = 4$; Trans-PyMT mice $n = 10$), DMXAA (Spont-PyMT mice $n = 3$; Trans-PyMT mice $n = 10$), and HBSS/ DMSO (Spont-PyMT mice $n = 4$; Trans-PyMT mice $n = 6$). The data are from three independent experiments for both tumor models. Source data are provided as a Source Data file

rapidly in Trans-PyMT tumors (Fig. 2b). In addition, the monocytes (CD11b[+] Ly6C[high] Ly6G[neg] F4/80[lo]) and CD8 T-cell infiltrates remained, respectively, two times and five times smaller in Spont-PyMT tumors compared with Trans-PyMT.

Altogether, these data indicated that DMXAA injection in Spont-PyMT tumors induced the recruitment of few immune cells, and had a much smaller impact on tumor regression than in the transplanted tumor model.

**Type I IFN production is not induced in Spont-PyMT tumors**. As DMXAA induced a weak immune response in Spont-PyMT mice, we measured the cytokines and chemokines expressed in the tumors of treated mice. DMXAA is known to activate the ubiquitous adaptor STING and subsequently triggers the production of TNFα, IFNα/β, and chemokines through NFκB, the phosphorylation of IRF3 (pIRF3), and STAT6, respectively[27,28]. Thus, we measured at the mRNA level the cytokines expressed in

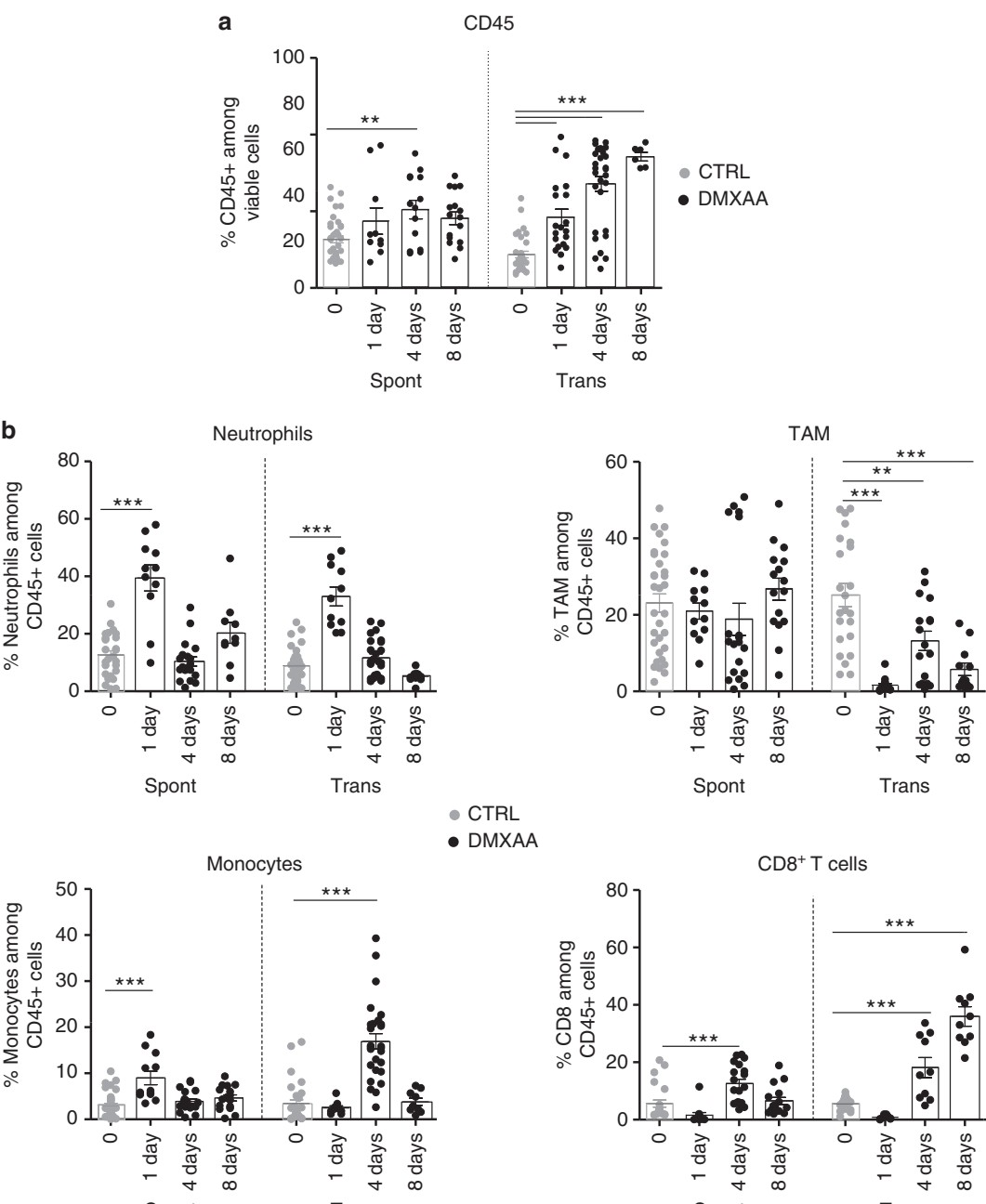

**Fig. 2** DMXAA induces a weak immune infiltrate in Spont-PyMT tumors. **a** The proportion of CD45+ cells in Spont-PyMT or Trans-PyMT tumors before treatment (gray circles) and at days 1 to 8 after DMXAA i.p. injection (black circles) was determined by flow cytometry after dissociation of the tumors. **b** The proportion of neutrophils, TAM, monocytes, and CD8+ T cells among those CD45+ cells are shown. For day 0, Spont-PyMT $n = 12$ and Trans-PyMT $n = 30$ mice, for day 1 Spont-PyMT $n = 6$ and Trans-PyMT $n = 12$, for day 4 Spont-PyMT $n = 9$ and Trans-PyMT $n = 24$ and for day 8 Spont-PyMT $n = 7$ and Trans-PyMT $n = 10$. The data are from four to seven independent experiments for each group of mice. Results are expressed as mean ± SEM. Tukey's multiple comparison test. *$p < 0.05$; **$p < 0.01$; ***$p < 0.001$. Source data are provided as a Source Data file

the tumor microenvironment shortly after DMXAA injection in Spont-PyMT mice. As shown in Fig. 3a, the transcription of *Tnfα*, *Cxcl1*, *Ccl2*, and *Ccl20* was upregulated in Spont-PyMT tumors, like in Trans-PyMT ones (Supplementary Fig. 2a), even though the fold increases were not always similar in the two tumor types. Strikingly, no significant increase in mRNA levels of *Ifnα* and *Ifnβ* genes was detected in the spontaneous tumor model (Fig. 3a), whereas these genes were highly expressed from 3 h to 24 h after DMXAA treatment in transplanted tumors (Supplementary Fig. 2a). Moreover, a positive feedback loop mediated by IFNαR

signaling normally allows further upregulation of *Irf7* mRNA and IFNα production. As expected, no upregulation of *Irf7* gene expression took place in Spont-PyMT tumors after DMXAA compared with Trans-PyMT tumors (Supplementary Fig. 2b).

We tested if this lack of type I IFN production after DMXAA injection in the spontaneous PyMT mice was DMXAA specific. To this aim, we used another STING agonist, the CDN 2′3′ cGAMP drug. The intratumoral injection of CDN 2′3′ cGAMP also failed to induce the upregulation of *Ifnα* or *Ifnβ* genes in Spont-PyMT tumors, whereas it did in Trans-PyMT tumors

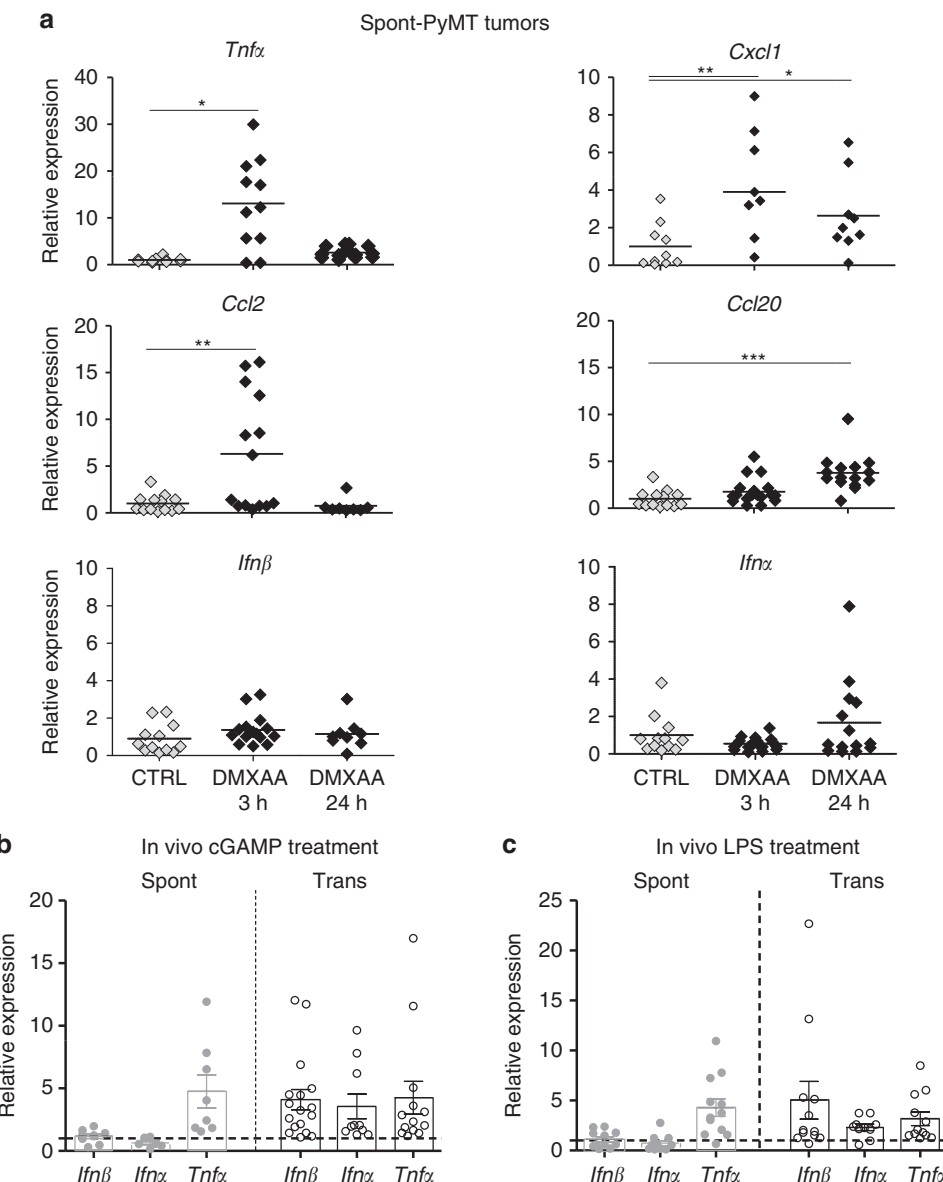

**Fig. 3** Lack of type I IFN triggering in Spont-PyMT tumors. **a** Spont-PyMT mice were injected i.p. with DMXAA and then killed after 3 h or 24 h to measure mRNA levels of cytokines and chemokines in tumors. The relative expressions in DMXAA-treated (black diamonds), compared with DMSO-treated (gray diamonds), mice are shown. Each dot corresponds to one tumor. Cumulative data from CTRL: $n = 6$ mice and DMXAA: $n = 6$ mice, from three independent experiments, are shown. Tukey's multiple comparison test. $*p < 0.05$; $**p < 0.01$; $***p < 0.001$. **b**, **c** The relative expression of these cytokines/chemokines, compared with PBS (CTRL) injected mice, were measured in Spont-PyMT (gray circles) and Trans-PyMT (open circles) target lesions 3 h after (**b**) cGAMP (25 μg i.t.) injection (Spont-PyMT mice: $n = 3$ CTRL and $n = 3$ cGAMP; Trans-PyMT mice: $n = 5$ CTRL and $n = 13$ cGAMP) or (**c**) LPS (50 μg i.p.) injection (Spont-PyMT mice: $n = 3$ CTRL and $n = 3$ LPS; Trans-PyMT mice: $n = 5$ CTRL and $n = 10$ LPS) from three independent experiments. In **b** and **c**, the results are expressed as mean ± SEM. Source data are provided as a Source Data file

(Fig. 3b). Moreover, the TLR4 ligand LPS, another inducer of type I IFN, induced *Ifnα*, *Ifnβ*, and *Tnfα* mRNA levels within 3 h in Trans-PyMT tumors, while only *Tnfα* was upregulated by LPS in Spont-PyMT tumors (Fig. 3c).

Taken together, these data show that Spont-PyMT mice have an intrinsic defect in the production of type I IFN in response to STING or TLR4 stimulation, which may explain their resistance to DMXAA treatment.

**DMXAA fails to induce the phosphorylation of IRF3.** To identify at which molecular level the induction of type I IFN expression was blocked, we next analyzed the early signaling molecules involved in the STING activation pathway. The

phosphorylation of IRF3 (pIRF3) is necessary for type I IFN production. We first measured pIRF3 by immunofluorescence in tumor slices from transplanted and spontaneous tumors, as soon as 3 h after DMXAA injection. A strong pIRF3 labeling was detected after DMXAA treatment in slices of Trans-PyMT tumors (Fig. 4a, left panel), including in myeloid cells (F4/80[+]) and some endothelial (CD31[+]) cells (Fig. 4b). By contrast, pIRF3[+] cells were hardly detectable in tumor slices of DMXAA-treated Spont-PyMT tumors (Fig. 4a, right panel). Similar results were obtained after intratumoral injection of DMXAA or ML RR-S2 CDA (Supplementary Fig. 3). These results indicate that injection of STING agonists failed to activate the IFN pathway in any cell type of the tumor ecosystem in Spont-PyMT mice, due to a

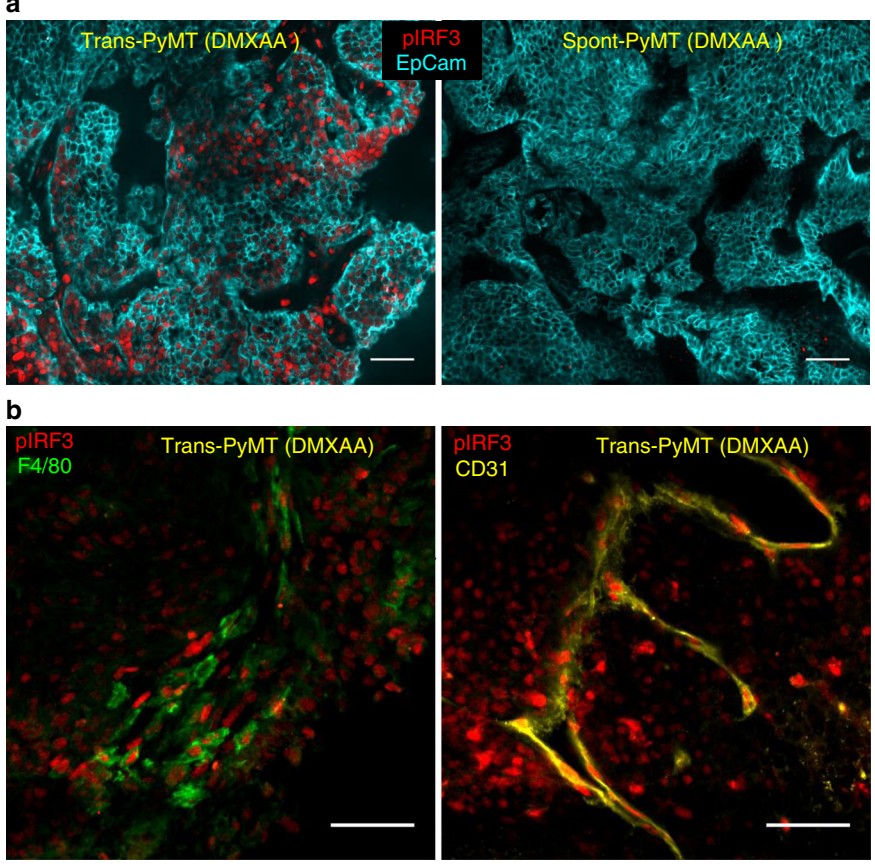

**Fig. 4** Absence of IRF3 phosphorylation in Spont-PyMT tumors after DMXAA injection. **a** Spont-PyMT mice and Trans-PyMT mice received one i.p. injection of DMXAA. Three hours later, tumors were collected and tumors slices were stained with anti-EpCAM (blue) and anti-pIRF3 (red) mAbs. Representative images from Trans-PyMT tumors (left) and Spont-PyMT tumors (right) are shown. **b** In Trans-PyMT tumors, some pIRF3+ cells (red) are myeloid (F4/80+, green), or endothelial cells (CD31+, yellow). Scale bars = 50 μm. The data are representative of three Spont-PyMT and five Trans-PyMT mice, from five independent experiments

blunted pIRF3 level. We verified that there was no difference in *Sting*, *Irf3*, and *Irf7* intratumoral mRNA transcripts between Spont- and Trans-PyMT mice (Supplementary Fig. 4a). In addition, the Tank-binding kinase 1 (TBK1), which is responsible for the phosphorylation of IRF3, was phosphorylated in both tumor models after DMXAA treatment (Supplementary Fig. 4b). These results are consistent with the upregulation of *Tnf*α and chemokines mRNA levels after DMXAA injection (Fig. 3a) and indicates that the signaling pathway leading to type I IFN production is altered downstream of TBK1 activation.

Taken together, these results indicate that, although the two models were based on the same tumor cells, the tumor microenvironment of Spont-PyMT mice is resistant to the induction of type I IFN production. This defect is likely due to the blockade of IRF3 phosphorylation.

**Anti-TGFβ unlocks type I IFN production by macrophages**. TGFβ is frequently accumulated in tumors where it is known to inhibit the production of pro-inflammatory cytokines by macrophages, including IFNβ[29]. Therefore, we examined if TGFβ was involved in the impaired type I IFN production following DMXAA administration to Spont-PyMT mice. Strikingly, *TGFβ* mRNA was expressed at much higher levels (~sevenfold) in untreated Spont-PyMT tumors than in Trans-PyMT ones (Fig. 5a). In addition, as shown in Fig. 5b, the vast majority of cells in spontaneous tumors, and not in transplanted ones, showed nuclear pSmad2/3 expression, indicating active signaling

through TGFβR in these tumors. Nuclear pSmad2/3 was found in F4/80+ myeloid cells, gp38+ fibroblasts, and EpCAM+ tumor cells (Fig. 5c).

We thus wondered if an anti-TGFβ treatment could render Spont-PyMT mice more sensitive to DMXAA. To test this, Spont-PyMT mice were treated with an anti-TGFβ antibody for 4 days then injected with DMXAA, killed 3 h later, and the phosphorylation of IRF3 was measured in tumor slices. As shown in Fig. 6a, the pretreatment of mice with an anti-TGFβ in vivo allowed DMXAA to induce the phosphorylation of IRF3 in a fraction of cells. This is in contrast with the total absence of DMXAA-induced pIRF3 when Spont-PyMT are not treated with an anti-TGFβ, as shown in Fig. 4a. Among the pIRF3+ cells, both F4/80+ myeloid cells and tumor cells were identified (Supplementary Fig. 5a). Moreover, this anti-TGFβ neutralization unleashed the expression of *Ifn*α and *Ifn*β genes after DMXAA injection in Spont-PyMT treated mice (Supplementary Fig. 5b).

To check the importance of myeloid cells in the initial DMXAA-induced burst of inflammatory cytokines[16,30], we depleted these cells in vivo before DMXAA injection. This experiment demonstrates that myeloid cells were indeed responsible for type I IFN expression (Supplementary Fig. 5b). We further analyzed the proportion and phenotype of myeloid cells infiltrating the anti-TGFβ-treated Spont-PyMT tumors. There was no quantitative difference in terms of cell proportions between anti-TGFβ-treated and untreated Spont-PyMT tumors. In both cases, TAM was the main myeloid cell population of the immune infiltrate (Supplementary Fig. 5c). However, there was a

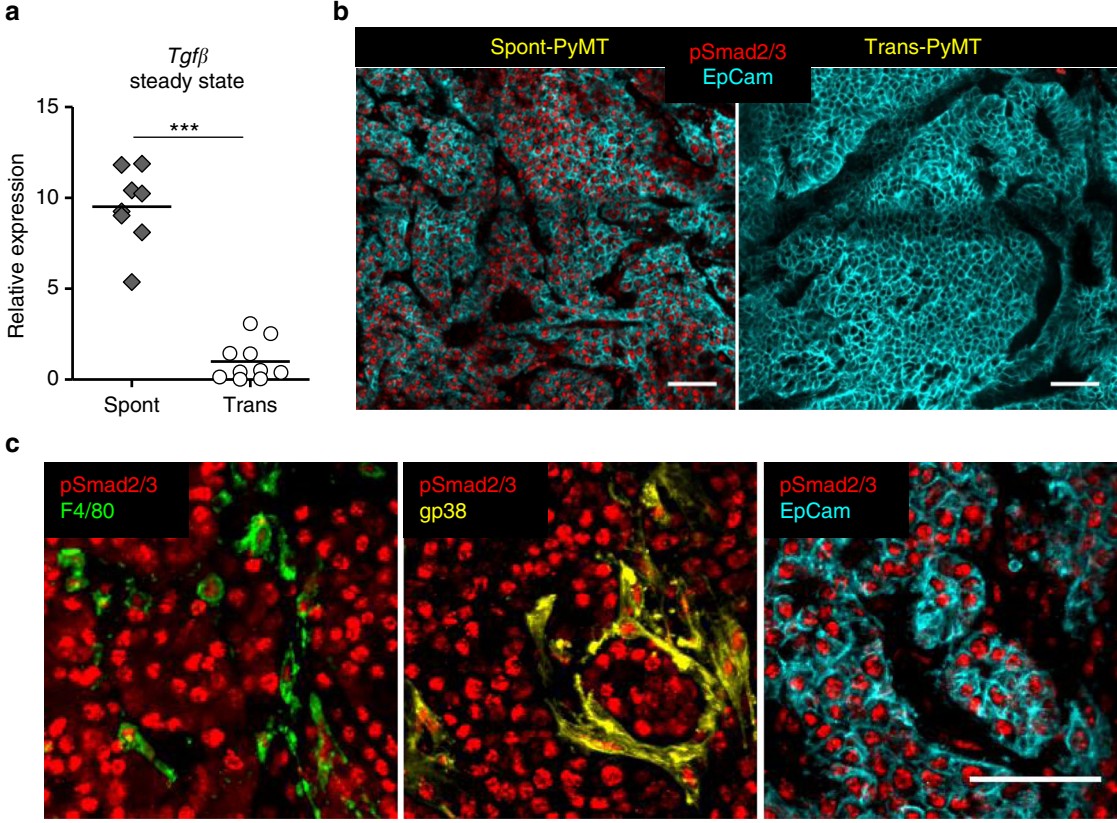

**Fig. 5** TGFβ expression is associated with pSmad2/3 in Spont-PyMT tumors. **a** mRNA levels of *Tgf*β were measured in untreated Spont-PyMT (gray diamonds) and Trans-PyMT (open circles) tumors, right after tumor dissociation. The relative expression, normalized to Trans-PyMT tumors, is shown. Each dot corresponds to one tumor. Cumulative data from Trans-PyMT mice: $n = 10$ and Spont-PyMT mice: $n = 3$. Student's $t$ test. $*p < 0.05$; $**p < 0.01$; $***p < 0.001$. **b** Tumors slices from Spont- and Trans-PyMT-untreated mice were stained with anti-EpCAM (blue) and anti-pSmad2/3 (red) Abs. **c** In Spont-PyMT tumors, pSmad2/3 + cells (red) may be myeloid cells (F4/80+, green), fibroblasts (gp38+, yellow), or tumor cells (EpCAM+, blue). Note that the 2nd and 3rd images, not superimposed for clarity, are from the same image field. Images are representative of Spont-PyMT: $n = 3$ mice; Trans-PyMT: $n = 3$ mice, from three independent experiments for each group of mice. In **b** and **c**, scale bars = 50 μm. Source data are provided as a Source Data file

qualitative difference in these cells. Indeed, TGFβ blockade was accompanied by a doubling in activated MHCII+ TAM compared with untreated mice, reaching frequencies comparable with those observed in Trans-PyMT tumors (Fig. 6b). The few infiltrating monocytes were also mainly MHCII+ in anti-TGFβ-treated mice as in Trans-PyMT. Finally, we purified the immune cells from anti-TGFβ-treated Spont-PyMT tumors and tested if F4/80+ cells were responsible for IFNα production. After in vitro stimulation with DMXAA, the F4/80+ cell fraction produced a much larger amount of IFNα than the F4/80neg fraction (Fig. 6c). Furthermore, when MHCII+ subsets of TAM were sorted and stimulated in vitro with DMXAA, they tended to produce more IFNα than their MHCIIneg counterpart (Supplementary Fig. 5d).

Together, these results indicated that TGFβ blockade rendered Spont-PyMT mice more sensitive to DMXAA stimulation by promoting the infiltration of mammary tumors by activated MHCII+ TAM capable of producing type I IFN.

**TGFβ inhibits type I IFN production by macrophages**. To decipher by which molecular mechanisms TGFβ could interfere with the production of type I IFN in macrophages, we switched to in vitro experiments performed with bone marrow-derived macrophages (BMDM). After 1 week of differentiation, BMDM were exposed to TGFβ overnight, and their capacity to phosphorylate IRF3 in response to DMXAA was evaluated. We found that TGFβ-treated BMDM had strikingly lost their ability to phosphorylate IRF3 when stimulated with DMXAA for 3 h

(Supplementary Fig. 6a, b). It has been reported that the histone deacetyl transferase HDAC4 in the cytoplasm can inhibit type I IFN production[31] and that TGFβ can stimulate a ROS-dependent export of HDAC4 out of the nucleus[32]. We thus examined whether TGFβ signaling could also alter the subcellular localization of HDAC4 in macrophages. In BMDM, HDAC4 was mainly nuclear (Supplementary Fig. 6c). By contrast, exposure of BMDM to TGFβ overnight induced a substantial translocation of the molecule to the cytoplasm, as shown by its homogeneous distribution in these cells. Moreover, the inhibition of ROS production during TGFβ exposure decreased this translocation of HDAC4 (Supplementary Fig. 6c) and restored the phosphorylation of IRF3 after DMXAA stimulation (Supplementary Fig. 6a, b).

Together, these results showed that TGFβ could have an indirect effect on DMXAA-induced IFN expression by triggering molecular changes in macrophages susceptible to affect their ability to produce type I IFN. We identified ROS-mediated relocalization of HDAC4 as one molecular mechanism by which TGFβ can specifically interfere with the phosphorylation of IRF3 and thus regulate type I IFN response in these cells.

**Blocking TGFβ unleashes DMXAA-induced tumor regression**. We finally wondered if TGFβ blockade could facilitate DMXAA-induced tumor regression in Spont-PyMT mice. Mice were treated with a combination of anti-TGFβ and DMXAA, the latter being injected once i.p., 4 days after the beginning of the anti-

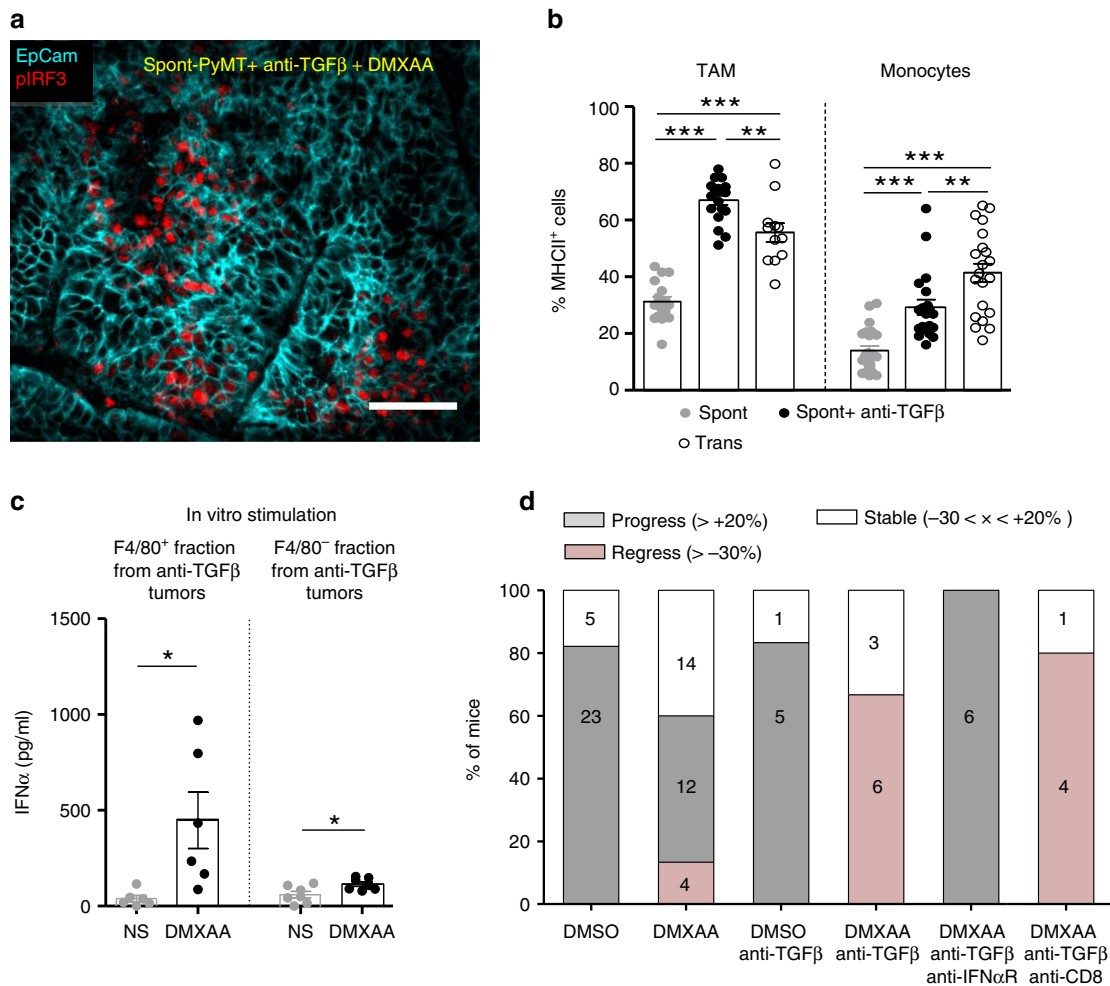

**Fig. 6 Blocking TGFβ allows DMXAA-induced type I IFN production and tumor regression. a** Spont-PyMT mice, treated with an anti-TGFβ (day − 4 and day − 1), received one i.p. injection of DMXAA. Three hours later, tumors were collected and tumor slices were stained with anti-EpCAM (blue) and anti-pIRF3 (red) mAbs. The image is representative of $n = 4$ anti-TGFβ-treated Spont-PyMT mice from three independent experiments. Scale bar = 50 μm. **b** Tumors from Spont-PyMT mice, treated with anti-TGFβ (black circles) or notβ (gray circles), and from Trans-PyMT mice (open circles), were analyzed by flow cytometry for the expression of MHCII on TAM and monocytes. The data are from anti-TGFβ-treated Spont-PyMT mice: $n = 7$; Spont-PyMT mice: $n = 6$ or 8; Trans-PyMT mice: $n = 12$ or 21, for TAM or monocytes respectively, collected in seven independent experiments. The results are expressed as mean ± SEM. Tukey's multiple comparison test. *$p < 0.05$; **$p < 0.01$; ***$p < 0.001$. **c** Tumors from anti-TGFβ-treated Spont-PyMT mice were pooled for each mouse (8–10 tumors), dissociated and F4/80$^+$ cells were sorted with magnetic beads. F4/80$^+$ and F4/80$^{neg}$ fractions were restimulated in vitro with DMXAA (250 μg/ml, black circles) or not (gray circles) and IFNα was measured by ELISA in the supernatant after overnight culture. The results are expressed as mean ± SEM. Student $t$ test. *$p < 0.05$; **$p < 0.01$; ***$p < 0.001$. **d** Spont-PyMT mice were treated with DMXAA (i.p. at day 0) or DMSO as a control, with anti-TGFβ (i.p. 200 μg), anti-IFNαR (i.p. 200 μg) and/or with anti-CD8 (i.p. 200 μg) Abs on day − 4, day − 1, day 1, day 4. The proportion of mice with an average regressing, stabilized or progressing tumors, is shown as in Fig. 1. DMSO-treated mice: $n = 28$; DMXAA-treated mice: $n = 30$; DMSO + anti-TGFβ: $n = 6$; DMXAA + anti-TGFβ: $n = 9$; DMXAA + anti-TGFβ + anti-IFNαR: $n = 6$; DMXAA + anti-TGFβ + anti-CD8: $n = 5$. Statistics are shown in Supplementary Table 2. Source data are provided as a Source Data file

TGFβ treatment. In these conditions, the tumor burden of Spont-PyMT mice was diminished (in 66%) or stabilized (in 33%) within 5 days following the combined treatment (Fig. 6d). This response was not dependent on CD8 T cells, but required IFN signaling since it was completely blocked in mice treated with an anti-IFNAR antibody (Fig. 6d). Altogether, these results showed that blocking TGFβ allowed DMXAA-induced type I IFN production by tumor-infiltrating macrophages and facilitated a transient tumor regression in Spont-PyMT mice.

## Discussion

Type I IFN have the unique potential to promote the activation and recruitment of various immune effectors and represent a natural defense, mainly against viruses. IFN can also help to fight cancer and encouraging results have already been obtained in transplanted tumor models with STING ligands, able to trigger IFNα/β production[10,16]. However, whether one can induce the production of type I IFN therapeutically in spontaneous tumors remains a burning issue. Here, we provide an example in which TGFβ accumulation in the microenvironment of spontaneous tumors blocks the production of IFNα/β by macrophages after STING activation and contributes to prevent tumor rejection. This obstacle can be overcome with an anti-TGFβ treatment.

In sharp contrast with the tumor regression systematically induced by DMXAA in transplanted PyMT mice[16], we report here that Spont-PyMT tumors rarely regress after STING

activation. A weak immune response was associated with the failure of DMXAA-induced tumor regression. What was specifically lacking in spontaneous tumors was the production of IFNβ and IFNα, not the production of immune-attractant chemokines. Four distinct type I IFN inducers (DMXAA, LPS, cGAMP, and ML RR-S2 CDA) failed to induce the production of IFNα/β in these spontaneous tumors. These results suggest that spontaneous tumors develop some particular features that inhibit the induction of an IFN-dependent anti-tumor immunity, a hypothesis which prompted us to study the activation of this pathway in the Spont-PyMT tumor model.

We found that the absence of type I IFN mRNA after DMXAA injection in Spont-PyMT mice was associated with a dramatic defect in IRF3 phosphorylation in the tumor ecosystem, both in tumor cells and in immune-infiltrating cells. On the contrary, the activation of TBK1, necessary for the activation of IRF3, was preserved, indicating a specific block downstream of the STING/TBK1 pathway. The transcription of the *Ifn*β gene, which is known to be driven by pIRF3 homodimers or by pIRF3/pIRF7 heterodimers, was thus likely blocked at an early stage. Moreover, no upregulation of *Irf7* gene expression took place in Spont-PyMT tumors after DMXAA compared with Trans-PyMT tumors, probably as a direct consequence of the absence of pIRF3 and IFNβ.

Decreased responsiveness to IFNα has already been reported in cancer patients[33], with reduced phosphorylation of STAT1 after stimulation with IFNα in vitro. Our study adds a key element: the capacity to produce IFNβ and IFNα may also be compromised in some tumors. This finding highlights that a defect in IFNα/β production in a tumor may originate from an early block in STING/IRF3 signaling. This adds up to the well-known defective IFNα production by some tumor cells, which may result from mutations or epigenetic regulation of this pathway[34,35], and which provides an advantage for therapeutic interventions with oncoviruses[35]. Here, the therapeutic potential of STING agonists to boost an anti-tumor response in solid tumors relies on the capacity of the tumor microenvironment to produce type I IFN. The Spont-PyMT mammary model illustrates that some tumors may be resistant to such interventions.

Our data further suggest that following DMXAA treatment, it is the tumor microenvironment that chiefly governed the IFNα/β production. Thus, endothelial cells and a large fraction of myeloid cells in Trans-PyMT tumors, strongly phosphorylated IRF3 following DMXAA injection. These results are consistent with earlier reports by Demaria et al. showing that endothelial cells rapidly produce IFNβ following injection of STING agonist[11]. Note, however, that in our experiments, endothelial cells were unlikely to be the main source of type I IFN as suggested in Demaria's work, but represented only a minute fraction of pIRF3+ cells, susceptible to produce IFNβ. Together, our data therefore suggest that the different abilities of Trans- and Spont-PyMT mice to produce type I IFN resides in different tumor microenvironments, essentially in myeloid cells.

We found that an anti-TGFβ treatment allowed DMXAA to induce the activation of IRF3, the production of IFNα/β, and to facilitate tumor regression in Spont-PyMT mice. TGFβ is fundamental in many physiological functions, including the regulation of mammary gland development. Although it may inhibit cell proliferation (and therefore, initial tumor growth), TGFβ is better known for its negative role later on. Indeed, it suppresses immune effector functions, accelerates metastasis dissemination[36], and this effect is amplified by the frequent mutations in the TGFβR signaling pathway allowing tumor cells to escape TGFβ antiproliferative effect. Not surprisingly, high levels of the active form of TGFβ have been associated with poor survival in several advanced cancers[37]. Of particular interest, the density of

pSmad2/3 in the stroma, downstream TGFβR signaling, has been associated with poor prognosis in non-small cell lung cancer[38]. In Spont-PyMT mice, the *Tgf*β mRNA level was high, and nuclear pSmad2/3 was observed in the vast majority of cells of the tumor ecosystem, indicating that the cellular source of TGFβ is unlikely to be unique. We have not attempted to identify the source of TGFβ, but it is well known that it may be secreted by multiple cell types, including macrophages[39] and apoptotic cells[40]. Such signs of massive TGFβ signaling were never observed in Trans-PyMT tumors.

We consider likely that in Spont-PyMT mice, and probably other spontaneous tumor mice, a major mode of action of TGFβ is indirect. Indeed, it has been shown that TGFβR signaling is involved in the suppression of pro-inflammatory cytokines and the polarization of myeloid cells[29]. Our study provides elements for understanding these inhibitory effects. Indeed, we have described a situation where, following the action of a STING ligand, TBK1 is phosphorylated, but not IRF3. The observation of a TGFβ-induced cytoplasmic relocalization of HDAC4 in BMDM, in a ROS-dependent way, is striking. It is consistent with that previously reported in fibroblasts[32]. HDAC4 could be a key player in TGFβ-induced effects. It has been shown that HDAC4 can exert a negative feedback loop in IFNβ signaling, by interacting with the TBK1/IKKε complex[31]. Such an HDAC-TBK1 interaction could explain how TGFβ introduced a block between TBK1 and IRF3 phosphorylation. Additional mechanisms may affect the responsiveness of TGFβ-treated macrophages, like an interference by Smad2 and Smad3 that can directly inhibit IRF3, resulting in reduced IFNα/β production[41]. In parallel to these molecular changes, we found that tumors that are sensitive to type I IFN induction, i.e., Trans-PyMT or anti-TGFβ-treated Spont-PyMT, were infiltrated by a majority of MHCII+ TAM that produce more type I IFN than the MHCII− TAM, which dominate in Spont-PyMT-resistant tumors. In line with this, the presence of TGFβ in triple negative human breast tumors has been associated with the functional weakness of infiltrating plasmacytoid DC for producing type I IFN[42]. Our findings indicate that TGFβ blockade allows the reprogramming of anti-tumoral infiltrating myeloid cells. Similarly, the blockade of TGFβ signaling has been shown to promote the emergence of anti-tumoral neutrophils[43].

Numerous studies have proposed explanations for the protumoral action of TGFβ. Many of them deal with the direct immunosuppressive effect of TGFβ (including their release by regulatory T cells) on T cells[44–46]. Others attribute it to the polarization of myeloid cells toward an immunosuppressive phenotype[29], or to its effect on cancer-associated fibroblasts[47,48], and to the ability of T cells to contact tumor cells, even though the underlying molecular mechanism of inhibition remained elusive. We propose here an additional molecular mechanism by which TGFβ may exert a protumoral effect in spontaneous tumors, through the blunting of IFNα/β production.

We have shown here that an anti-TGFβ treatment unlocks the production of type I IFN by tumor-infiltrating macrophages in response to DMXAA injected in Spont-PyMT mice. These treated mice are not completely equivalent to the transplanted ones, in terms of DMXAA-induced IRF3 phosphorylation and IFNα production, and there are obviously other differences between the two tumor models, which have no reason to all be TGFβ-dependent. Nevertheless, this combined treatment allows for a transient regression of the tumor by a mechanism independent of CD8 T cells. It is possible that the release of type I IFN, together with TNFα, has a direct effect on the tumor vasculature and the killing of tumor cells[49]. Tumor-infiltrating macrophages are known to play an important role in the production of these cytokines, and we have previously proven their implication,

together with that of CD8 T cells, in the DMXAA-induced regression of Trans-PyMT tumors[16]. We propose that anti-TGFβ combined with DMXAA treatment in Spont-PyMT mice induced an innate anti-tumor response. Finally, the DMXAA-induced regression of spontaneous tumors after anti-TGFβ treatment is transient, indicating that the initial innate responses would deserve being reinforced and prolonged by a treatment boosting the endogenous adaptive anti-tumor response.

The TGFβ/type I IFN interference highlighted in this study is likely to matter not only for STING agonists under active development, but also for chemotherapeutic agents like anthra-cyclins, for which tumor regression has been associated with a type I IFN signature[50]. In the case of tumors like breast tumors that poorly respond to anti-PD1/PDL1 checkpoint inhibitors and in which TGFβ and M2-like macrophages are abundant[21,36], there would be a solid rationale for using a triple combination of anti-TGFβ, anti-PD-1 and a third partner, such as STING agonist or chemotherapy.

## Methods

**Animal studies.** MMTV-PyMT transgenic mice are maintained at the Cochin Institute SPF animal facility by backcrossing on FvB/NCrl mice (Charles River laboratories). Transplanted PyMT mice were generated by transplantation in the mammary gland of 8-week-old FvB mice of $1 \times 10^6$ cells, freshly isolated from tumor cell suspensions prepared from MMTV-PyMT tumors by mechanical and enzymatic dissociation with DNase I (100 μg/ml, Roche), liberase (250 μg/ml, Roche), and hyaluronidase (1 μg/ml, Sigma)[16]. Animal care was performed in compliance with all relevant ethical regulations for animal testing and research of the Federation of European Laboratory Animal Science association. All procedures were approved by the French animal experimentation and ethic committee of Paris Descartes University (CEEA 34, 16-063). Sample sizes were chosen to assure reproducibility of the experiments in accordance with the replacement, reduction and refinement principles of animal ethics regulation.

Mice with tumors of 6 mm in diameter received a single i.p. injection of DMXAA (23 mg/kg in DMSO, #D5817 purchased from Sigma) or 100 μl of 50% DMSO in PBS as control. When specified, DMXAA was injected i.t. ($1 \times 500$ μg). LPS ($1 \times 50$ μg) was injected i.p., cGAMP (InvivoGen, $1 \times 25$ μg) was injected i.t., and mice were killed 3 h later. ML RR-S2 CDA (MedChem Express, $3 \times 50$ μg) was injected i.t. at day 0, 3, and 6. For antibody treatment, 200 μg of anti-IgG1 (#BE0083: clone MOPC-21) or IgG2a (#BE0089: clone 2A3) were used for isotype control and 200 μg of anti-TGFβ (#BE0057: clone 1D11.16.8), anti-IFNαR (#BE0241: clone Mar1-5A3), anti-CD8 (#BE0004-1: clone 53-6.72), or anti-GR1 (#BE0075: clone RB6-8C5) antibodies, all purchased from Bioxcell, were injected every 2 days from days −3 to day + 6 after DMXAA treatment. In some experiments, mice were fed with chow containing PLX3397 or control chow (Plexxikon Inc., from day −2 to day + 1) to deplete macrophages. The evolution of tumor size after DMXAA treatment was estimated as follows. For each tumor, its size at time $t$ has been calculated, relative to its size at d0, when DMXAA has been injected. For each mouse, the average was calculated for all its measurable tumors. Another representation of DMXAA-induced changes in tumor size starts with a snapshot of the change in tumor burden at day 5 post-DMXAA injection, measured as above, and pooled for all the mice of the cohort in three groups: global progression of the tumor burden when the increase in a mouse is > 20%; global regression if the tumor burden has decreased by > 30%, and stabilization in between.

**Multicolor flow cytometry.** Tumor cell suspensions ($4 \times 10^6$) prepared by mechanical and enzymatic dissociation[16] were stained in 96-wells round bottom plates with live/dead staining (Blue fluorescent reactive dye, #L34962 Invitrogen) during 20 min at room temperature. For flow-cytometry analysis and sorting, Fc receptors were blocked with anti-FcR (anti-CD16 and CD32, at 5 μg/ml, Biolegend #101339). After two washes in PBS 2% FCS, cells were stained with the following antibodies (all used at 1:100): anti-CD11b-BV421 (#562605), CD64-APC (#558539), CD11c-PeCy7 (#557401), TCRβ-BV605 (#562840) and CD4-BV711 (#563050) all purchased from BD Pharmingen; anti-CD45-AF700 (#103128), Ly6C-APCCy7 (#128025), Ly6G-BV510 (#127633), F4/80 BV650 (#123149), CD206-PE (#141706), IA/IE-BV785 (#107645) all purchased from Biolegend, and CD8-PerCPef710 (#46-0081-82) purchased from eBioscience. After washing, cells were fixed in 1% PFA, stored at 4 °C, and acquired the next day on LSR II or FORTESSA (BD Bioscience). For detection of phosphorylated proteins, cell suspensions were stimulated 3 h with DMXAA 250 μg/ml, fixed immediately in PFA 4%, permeabilized with frozen methanol 90%, stained overnight with 1:100 pTBK1-PE (#13498) antibodies (purchased from Cell signaling), then washed and further stained for multicolor flow cytometry.

**Differentiation and culture of bone marrow-derived macrophages (BMDM).** Bone marrow cells were collected from FvB mice. With a syringe, the marrow was flushed out with a needle of 25 G into a sterile Petri dish with DMEM. Then, the cells were centrifuged, resuspended in DMEM supplemented with glutamine, 30% L929 conditioned medium, 20% FCS and 2% P/S. BMDM were used in experiments after 5 days of culture. BMDM were incubated overnight at 37 °C with TGFβ (Peprotech; 5 ng/ml), with or without the ROS inhibitor NacetylCysteine (ThermoFisher; 1 nM) or left untreated. Then, cells were stimulated or not with DMXAA (Sigma; 250 μg/ml) during 3 h and analyzed by immunofluorescence.

**Immunofluorescence.** Tumor pieces were fixed overnight with periodate–lysine–paraformaldehyde at 4 °C. Then tumor samples were embedded in 5% low-gelling temperature agarose (type VII-A; Sigma-Aldrich) prepared in PBS. In total, 350-μm slices were cut with a vibratome (VT 1000 S; Leica) in a bath of ice-cold PBS[51]. Immunostaining of surface markers was performed at 37 °C for 15 min with antibodies specific for F4/80-AF488 (#MCA497A647) or AF647 (#MCA497A488) both purchased from Bio Rad, EpCAM-BV421 (#563214), CD31-Biot (#553371) both from BD Pharmingen or Gp38-PE (#127408) from biolegend (all used at 1:100). CD31-Biot was revealed with a streptavidin-PE (#554061) from BD Pharmingen. For additional intracellular staining, tissue slices were fixed in 4% PFA (10 min at room temperature) and permeabilized with methanol 90% 30 min at 4 °C and stained overnight with 1:10 pIRF3-AF647 (#103275) or 1:100 p-SMAD2/3 (#8685) both purchased from Cell signaling. For detection of pSMAD2/3, primary anti-pSMAD2/3 Ab was revealed with 1:200 of a goat anti-rabbit AF488 (#A11070) from life technologies.

Images of tumor slices were obtained using a confocal spinning-disk (CSU-X1; Yokogawa) upright microscope (DM6000FS; Leica) equipped with an ORCA Flash4.0LT camera (Hamamatsu) and a 25 × 0.95NA W objective (Leica). All images were acquired with MetaMorph 7 imaging software (Molecular Devices) and analyzed with Image J.

For the staining of BMDM, cells were fixed in PFA 4% immediately after the stimulation, permeabilized with frozen methanol 90%, stained overnight with 1:10 pIRF3-AF647 (#103275) from Cell signaling or 1:100 HDAC4 (#ab79521) from Abcam. For the detection of HDAC4, primary anti-HDAC4 Ab was revealed with an 1:200 goat anti-rabbit AF488 from life technologies. Images of BMDM were acquired on a wide-field Nikon Eclipse microscope through a ×20 objective, with an Eclipse TE2000, a cascade CDD camera (Photometrics). Images of pIRF3 had to be acquired with a binning of three, and five images were averaged. Images of HDAC4 were acquired without binning. For the quantification of pIRF3 intensity, performed on images of BMDM after background subtraction, the mean nuclear intensity was measured in 40–60 cells per condition.

**Transcriptomic analysis.** Tumor RNA was extracted from cut tumors using a RNeasy Mini Kit (Qiagen) according to the manufacturer's instructions. The RNA were reverse transcribed using the Advantage® RT for PCR kit (Applied Clontech), and gene expression was analyzed by RT-qPCR with the LighCycler® 480 Real-Time PCR system. The list of primers that were used is provided in the Supplementary Table 1.

**ELISA.** The ELISA for IFNα production (IFNα Mouse ELISA Kit from Thermo Fisher) was performed on cells purified from tumor cell suspensions after Ficoll density separation (Histopaque-1093 from Sigma) followed by purification with anti-F4/80 MicroBeads UltraPure kit (Miltenyi Biotec) to isolate F4/80[+] cells or followed by sorting of MHCII[+] and MHCII[neg] TAM subsets with ARIA III based on the gating displayed in the Supplementary Figure 1. Cells were then stimulated overnight with 250 μg/ml of DMXAA or left untreated before dosage of IFNα in culture supernatants.

**Statistics.** The data were analyzed with GraphPad Prism5 software to run unpaired Student's $t$ test or one-way ANOVA and Tukey test for multiple comparisons. For statistical analysis of in vivo treatments, the Kolmogorov–Smirnov test was run with Python software after estimation of the logarithmic fold change $\log_{10}(V_{tumor}$ [5 days]$/V_{tumor}[t=0])$ of the mean tumor volume in individual mice between day 0 and day 5 after treatment initiation. Values $\leq 0.05$ were considered significant. *$p < 0.05$; **$p < 0.01$; ***$p < 0.001$.

**Reporting summary.** Further information on research design is available in the Nature Research Reporting Summary linked to this article.

## Data availability

The authors declare that the data supporting the findings of this study are available within the article and its supplementary information files, or are available upon reasonable requests to the authors.

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

## Acknowledgements

We are really grateful to C. Kantari Mimoun for her critical review of the paper, and the staff of the IMAG'IC, CYBIO, and GENOM'IC facilities of the Cochin Institute for their advice all along this study. This work was granted by Worldwide Cancer Research (UK), the "Comité de Paris de La Ligue Contre le Cancer", the Cancer Research for Personalized Medicine (CARPEM), Plan Cancer (Tumor Heterogeneity and Ecosystem Program), the CNRS, INSERM, and University Paris Descartes. M. Guérin was granted by the Ministère de l'Education Nationale, de l'Enseignement Supérieur et de la Recherche. V. Finisguerra was granted by the "Fonds de la Recherche scientifique" (FNRS).

## Author contributions

N.B. and A.T. designed the study, and M.G. and N.B. designed the experiments. M.G., F.R., V.F., L.V., J.M.W., M.T., G.B., V.Fi., and N.B. performed and analyzed the experiments. T.G. helped with Image J analyses. G.A.B. helped with statistical analyses. M.G., A.T., and N.B. wrote the paper and prepared the figures. E.D. reviewed the paper.

## Additional information

**Competing interests:** The authors declare no competing interests.

