## [Peer Review File · Nature Communications]

Reviewers' comments:

Reviewer #1: Cancer immunotherapy
(Remarks to the Author):

The manuscript by Guerin and colleagues examines the effects of the STING agonist DMXAA in a genetically engineered mouse model of breast cancer, compared to the same treatment in a transplantable version of the same tumor. They find that a single injection of DMXAA has very limited efficacy in the GEM model. This is associated with minimal induction of pIRF3 and minimal type I IFN production, which was correlated with high TGF- β signaling and pSMADs in the GEM model versus the transplantable tumor. Blockade of TGF- β could improve immune infiltration and therapeutic efficacy of DMXAA. These are important results that have implications for clinical translation, given the prevalence of high TGF- β expression and signaling in human cancers. Nonetheless, there are critical mechanistic details not addressed that would increase the impact of the current manuscript.

Specific comments:

1. The difference between the GEM tumors and transplantable tumors are interesting and noteworthy. However, two aspects are unclear based on the way STING agonists are being utilized in the clinic. First, the DMXAA appears to have been given i.p.—this is not evident from the Results section or the figure legend but is only described in the Methods—this fact should be stated up front in the Results. STING agonists are generally being investigated with intratumoral injection (in order to focus immune priming on tumor antigen-specific T cells). It is possible that the drug distribution is distinct in the GEM model compared to the transplantable model, or that the vascular endothelial cells are differentially affected from the endovascular side. Therefore, it is important to assess intratumoral injection in both models. Second, it appears that only a single injection has been given. Clinically, STING agonists are being injected along a regular schedule, and some mouse models have also utilized repeated injections. The current model also should explore multiple injections. The bottom line is that it has not been adequately demonstrated that DMXAA is inactive in the GEM model context, especially given the fact that the single systemic injection did indeed slow tumor growth.
2. The mechanism by which TGF- β mediates blunted type I IFN production in the GEM model is not clear. Is it through altering the differentiation state of myeloid cells (and/or other cells) in the tumor microenvironment, or is there a cross-talk on signaling within individual cells between the TGF- β and STING/IRF3 pathways? If the latter, what is the biochemical/molecular mechanism by which TGF- β inhibits pIRF3 without affecting TBK1 activation? This could be explored with defined cell lines in vitro. Closing the mechanistic loop would increase the impact of the data.
3. The mechanism for improved tumor regression in the GEM model with DMXAA+anti-TGF- β is not clear. Does it depend on T cells? Does it depend on type I IFNs? Does it lead to increased priming of tumor-specific immune cells? Is T cell trafficking improved? Understanding the cellular mechanisms of this combinatorial therapy also would increase the impact of the work, especially as STING agonists are being explored clinically in cancer patients.
4. p4: There is no need to editorialize that "Ming et al nicely described...". It is preferable just to state the data.

Reviewer #2 : Breast Cancer and microenvironment
(Remarks to the Author):

This study by Gurin et al. purport to show that TGF β blockade is a key to DMXAA-induced tumor regression by restoring the production of IFN α following STING activation to enable tumor regression. One model MMTV-PyMT is used and the DMXAA efficacy in the tumors arising from transplantation of primary tumor cells is compared to that in spontaneous tumors.

The MMTV-PYMT model develops palpable mammary tumors as early as 5 weeks of age that are

multifocal, highly fibrotic, and involve the entire mammary fat pad. Moreover, they metastasize to the lung almost concomitant with palpation. Thus tumor burden is an uncontrolled variable in the comparison to a single transplanted tumor. Indeed given this aspect of the spontaneous model, it is important for the authors to indicate total tumor burden and explicitly state what %change means in Figure 1 growth response (e.g. index lesion, per palpable tumor, total tumor volume). Indeed, it is only at the end of the last figure legend that the authors state that "Data are derived from 3-5 mice from 3 independent experiments." Indicating that tumors growth is not independent measurements (i.e. within same mouse). This is a major limitation of the study, especially in light of the introduction in which the authors lead the reader to consider the evolving tumor microenvironment, which is an important determinant, but is not controlled for in this study in which a mouse with a single tumor is compared to mice with multiple tumors and metastatic disease. This aspect of the paper should be reviewed by a statistician.

Throughout the manuscript, the experimental approaches are not self-evident as the figures move from in vivo to in vitro, injection to stimulation, protein to mRNA without explanation leaving the reader to puzzle out by reference to methods and figure legends what was done, which is still often unclear. In some cases, the methods are in the figure legends (e.g. Fig 3C legend), rather than organized in the methods.

Similarly, in the FACS analysis, immune infiltrate is reported as % viable cells, which includes tumor cells. Thus changes do not reflect the immune infiltrate per se but rather are confounded by the growth rate of the tumor, clearly different in the spontaneous and transplanted tumors. Infiltrate would be more informative if normalized to CD45+ cells. In this regard, immune cell phenotypes are presented after the fact so to speak, for example in Figure 3, the authors conclude that DMXAA does not stimulate IRF3 phosphorylation in tumor cells and immune infiltrates in Spont-PyMT tumors but then go on to state the monocytes population are distinct, which is the fundamental basis for the difference. Given that TGF β is well established mediator of monocyte differentiation and function, the next experiment to show that its inhibition shifts macrophage polarization is predictable, but does not support the title "TGF β blocks STING-induced IFN α / β r..." as what TGF β does is to shift the myeloid population. Also the legend for figure 4 "The accumulation of TGF β in Spont-PyMT but not in Trans-PyMT tumors impaired the activation of IRF3 and the production of IFN after DMXAA activation" is not supported by the data presented therein.

In Figure 5, the authors conclude that "the neutralization of TGF β in vivo allowed the DMXAA-induced phosphorylation of IRF3" but given the 5 day pretreatment and the resulting shift in polarization, this is misdirection, that is then the basis for demonstration of the expected effect. Finally, the penultimate experiment of DMXAA +anti-TGF β lacks controls (i.e. TGF β neutralizing antibody alone, and irrelevant control antibody) as well as test against the transplanted tumor. Also the experiment is only carried out 5 days post treatment, which is insufficient to assess immunological mediated regression. Note too that the same degree of regression is seen in DMXAA treated alone waterfall plot, but fewer tumors did so.

Finally, the authors do not actually test their fundamental conclusion that STING is the essential mediator of tumor regression, nor that the difference between spontaneous and transplanted tumors is dependent on it.

Minor Points

Pg. 8, please state the route of LPS injection.

Fig. 2 Legend, please state how genes were measured and on what type of preparation. Also here, IFN γ was measured following in vitro stimulation, but this is not stated in text, and the methods indicate two different preps were used, total and sorted cells, although it is not stated on what marker the cells were sorted.

Point-by-point response to referee's comments

Specific comments from Reviewer #1:

1. The difference between the GEM tumors and transplantable tumors are interesting and noteworthy. However, two aspects are unclear based on the way STING agonists are being utilized in the clinic. First, the DMXAA appears to have been given i.p.—this is not evident from the Results section or the figure legend but is only described in the Methods—this fact should be stated up front in the Results. STING agonists are generally being investigated with intratumoral injection (in order to focus immune priming on tumor antigen-specific T cells). It is possible that the drug distribution is distinct in the GEM model compared to the transplantable model, or that the vascular endothelial cells are differentially affected from the endovascular side. Therefore, it is important to assess intratumoral injection in both models. Second, it appears that only a single injection has been given. Clinically, STING agonists are being injected along a regular schedule, and some mouse models have also utilized repeated injections. The current model also should explore multiple injections. The bottom line is that it has not been adequately demonstrated that DMXAA is inactive in the GEM model context, especially given the fact that the single systemic injection did indeed slow tumor growth.

We agree with the reviewer that intratumoral (i.t.) injection of DMXAA would be a good option to rule out potential differences in drug distribution by systemic route between the two models. Before developing this point, we would like to emphasize that DMXAA is not inactive in the GEM model when injected i.p: indeed, we had shown that DMXAA induced the production of both TNF α and chemokines in the tumor (Fig. 2a) and the recruitment of a few immune cells (Fig. 1c and d) in these Spont-PyMT mice. The key point is that it specifically failed at inducing the phosphorylation of IRF3 and the production of type I IFN (Fig. 2a and Fig. 3b), this cytokine being key for an optimal immune cell recruitment and tumor regression, as we have shown earlier (*Weiss et al., Oncoimmunol 2017*). This phenomenon was also observed when the STING agonist, cGAMP, was injected i.t. as shown initially as supplementary data (now moved to Figure 2b in the revised version).

To follow the suggestions of the reviewer, we performed additional experiments to assess whether i.t. injection or multiple injections can bypass this specific blockade. Mice, either transplanted or GEM Spont-PyMT, received three i.t. injections of DMXAA or ML RR-S2 CDA at a 3 day interval, or saline buffer as control vehicle. DMXAA was injected i.t. at a dose of 500 μ g (*Corrales et al., Cell Reports 2015*), which is close to the dose injected i.p. in our previous experiments (23mg/kg in FVB mice of 22 to 24g). We also performed i.t. injections of the agonist ML RR-S2 CDA that has been shown to induce efficient tumor regression in previous transplanted tumor models with the dose of 50 μ g (*Corrales et al., Cell Reports 2015*). Mice were either sacrificed 3h after the last injection, to stain for pIRF3 in the tumor mass, or were followed for 20 days to measure the impact of the

treatment on the tumor growth. The new results show that, after multiple i.t. injections, both STING agonists failed to induce the phosphorylation of IRF3 and the regression of the spont-PyMT tumors, in sharp contrast with their effects on trans-PyMT tumors. **We have included these results in the revised manuscript (page 9 and Figure S3).**

We have further tested if the addition of DMXAA directly on freshly isolated slices of Spont-PyMT tumors could induce the phosphorylation of IRF3 (to make sure that the whole tumor surface was in contact with the drug) but again, it failed to induce pIRF3, as shown for the reviewers' courtesy in the figure 1 below (left panel), in contrast to tumors from anti-TGF β treated Spont-PyMT mice (right panel). Overall, these results show that the activation of IFN pathway is blocked in the Spont-PyMT tumors.

Figure1: The addition of the DMXAA drug directly on freshly isolated tumor slices failed to induce the phosphorylation of IRF3 in Spont-PyMT tumors (left) unless mice were pretreated with an anti-TGF β antibody (right). Fresh tumors from Spont-PyMT mice pretreated *in vivo* with anti-TGF β or IgG antibody (day-4 to day-1), were collected (day0), embedded in agarose and tumor slices of 300 μ m were further incubated for 1h at 37°C with DMXAA (250 μ g). Slices were then fixed, permeabilized and stained with EpCAM mAb (green) to delineate the structure of the tumor and with pIRF3 Ab (red).

2. The mechanism by which TGF β mediates blunted type I IFN production in the GEM model is not clear. Is it through altering the differentiation state of myeloid cells (and/or other cells) in the tumor microenvironment, or is there a cross-talk on signaling within individual cells between the TGF β and STING/IRF3 pathways? If the latter, what is the biochemical/molecular mechanism by which TGF-b inhibits pIRF3 without affecting TBK1 activation? This could be explored with defined cell lines *in vitro*. Closing the mechanistic loop would increase the impact of the data.

We fully agree with the reviewer that the mechanism by which TGF β blocks type I IFN production is intriguing and deserved further investigations. In the original version of the manuscript, we only showed that an anti-TGF β treatment was accompanied by the infiltration of activated MHCII⁺ tumor-associated macrophages and monocytes in Spont PyMT tumors, suggesting that TGF β was altering the differentiation status of the myeloid cells.

To further investigate how TGF β blocks the production of type I IFN at the molecular level, we made Bone-Marrow-Derived Macrophages (BMDM) from FVB mice, after one week of *in vitro* differentiation. We first investigated if we could reproduce *in vitro* the blunted type I IFN production observed in Spont-PyMT model. We found that, contrary to untreated BMDM, BMDM exposed to TGF β overnight were unable to phosphorylate IRF3 and to produce IFN α when stimulated with DMXAA (**Figure S5 and Figure 2a below**). Similar results were obtained when BMDM were exposed to TGF β for 3 days, but not when TGF β was added at the time of DMXAA stimulation, clearly suggesting an indirect effect of TGF β signaling.

To investigate the mechanism underlying this phenomenon, we first tested if a degradation of IRF3 protein in macrophages pretreated with TGF β could explain the decrease in pIRF3 after stimulation with DMXAA. This is not the case, since the total amount of IRF3 protein was similar in TGF β -treated or control BMDM, stimulated or not with DMXAA (**Figure 2b below**).

Figure 2: Exposure to TGF β alters the capacity of BMDM to express IFN α after DMXAA stimulation. BMDM were cultured overnight with TGF β , TGF β + ROS inhibitor or left untreated. Cells were then stimulated with DMXAA for 3 hours. **(a)** The relative expression of IFN α mRNA measured by qRT-PCR is shown, normalized to untreated BMDM. Mean \pm s.e.m. of 6 mice from 4 independent experiments. **(b)** Total amount of IRF3 and actin proteins were measured by western blot.

Based on two previous reports (*Yang et al., J Mol Cell Biol 2018; Guo et al., Am J Physiol lung cell mol physiol 2017*), we hypothesized that TGF β -induced HDAC4 translocation to the cytoplasm could be an intermediate step preventing type I IFN production. We found indeed that sustained TGF β signaling in BMDM leads to the nuclear export of HDAC4 to the cytoplasm, and interferes with the phosphorylation of IRF3 in response to DMXAA (Figure S5c). We could also show that (consistently with the reported role of ROS in TGF β -induced HDAC4 translocation), the inhibition of ROS production after a prolonged TGF β exposure blunted the translocation of HDAC4 out of the nucleus (Figure S5c), and partially restored the phosphorylation of IRF3 and the expression of IFN α after DMXAA stimulation (Figure S5a and b; and Figure 2a above). Together, our interpretation is that TGF β signaling in macrophages induces many molecular changes (or "*polarization*"). We have identified HDAC4-mediated interference with IRF3 phosphorylation as one mechanism by which TGF β can specifically negatively regulate type I IFN response at the molecular level in these cells.

In the revised version of the manuscript, we have included these important data in the results section (pages 10, 11 and Figure S5) and in the discussion (pages 15, 16). We are grateful to the reviewer for his/her question, which allowed us to provide answers that strengthen our conclusions regarding the effect of TGF β on the IFN pathway in the tumor microenvironment.

3. The mechanism for improved tumor regression in the GEM model with DMXAA+anti-TGF- β is not clear. Does it depend on T cells? Does it depend on type I IFNs? Does it lead to increased priming of tumor-specific immune cells? Is T cell trafficking improved? Understanding the cellular mechanisms of this combinatorial therapy also would increase the impact of the work, especially as STING agonists are being explored clinically in cancer patients.

We thank the reviewer for these questions. We have conducted 2 sets of experiments to address them:

a) Spont-PyMT mice were treated with monoclonal antibodies to block IFN α Receptor, in combination with DMXAA + anti-TGF β treatment. In these conditions, no tumor regression was observed. This shows that the pretreatment with anti-TGF β entails an IFN-dependent tumor regression (**Figure 3a below**). Strikingly, CD8 T cells were dispensable for this phenomenon, since an anti-CD8 depleting mAb did not prevent the transient tumor regression induced by the combined treatment (**Figure 3b below**).

Figure 3: The tumor regression induced in Spont-PyMT mice by the combined DMXAA + anti-TGFβ treatment requires (a) IFN but (b) not CD8 T cells. Spont-PyMT mice were treated with anti-TGFβ alone, anti-TGFβ + anti-IFNAR, anti-TGFβ + anti-CD8 or isotype control mAbs (day-4, day-1, day1, day5 and day9) and received one i.p. injection of DMXAA or vehicle (day0). The average relative changes in tumor size are shown for each group (DMXAA + IgG (n= 5 mice) and DMXAA + aTGFβ (n=2) from 2 to 3 independent experiments; DMXAA + aTGFβ + anti-IFNAR (n=6) and DMXAA + aTGFβ + anti-CD8 (n=5) from 3 independent experiments (with average changes of 8-10 tumors for each mouse). The dotted lines indicate the threshold for regression (reduction >30%) and the baseline level.

b) Despite the fact that the DMXAA-induced regression was CD8 T cell independent in this tumor model, we have further analyzed the levels of CD8 and CD4 T cell activation in the tumor draining lymph nodes (TDLN) of Spont-PyMT mice, in case the anti-TGFβ treatment would have affected the priming of T cells. Of note, DMXAA injected i.p. can potentially induce the release of multiple tumor-associated antigens following tumor cells death. Thus, the priming of a large repertoire of anti-tumor CD4 and CD8 T cells of unknown antigen-specificity could potentially take place in the tumor draining lymph node (TDLN) in this model. Obviously, in such settings, one cannot identify tumor-specific T cells, but we have looked at several T-cell activation markers (CD69, CD25, CD62L, CD44, PD1). We collected the TDLN, spleen and the tumors of Spont-PyMT mice 3 days after DMXAA +/- anti-TGFβ treatment. As positive controls, we used DMXAA-treated mice bearing transplanted PyMT tumors. We did not find any clear evidence for a better priming of TDLN T cells in anti-TGFβ treated animals. CD69 was the only marker for which we found a modest increase in the proportion of activated CD8 T cells in the tumor-draining lymph nodes (mean of 45% versus 32% of CD69⁺ CD8⁺ T cells in anti-TGFβ + DMXAA and DMXAA treated Spont-PyMT mice, p=0.03 student *t* test). These results are displayed in **Figure 4 below**. For comparison, DMXAA induced the upregulation of CD69 on 60% of the CD4 and CD8 T cells in TDLN and spleen of trans-PYMT mice. In addition, the proportion of

activated CD8 T cells in the tumor was also similar in DMXAA-injected Spont-PyMT mice, treated or not with anti-TGFβ.

Figure 4: Anti-TGFβ treatment did not increase the priming of CD8 T cells or CD4 T cells in response to DMXAA in Spont-PyMT compared to DMXAA treatment alone. Spont-PyMT mice (a and c) were treated with DMSO (ctrl), DMXAA (black) or DMXAA + anti-TGFβ (blue). Transplanted PyMT mice (b and d) were treated with DMXAA as positive controls for an optimal T cell activation following DMXAA treatment. After 3 days, tumor draining lymph nodes, tumors and spleen were collected and the proportion of CD69⁺ CD8 T cells (Top) or CD4 T cells (bottom) were analyzed by flow cytometry. Data represent the mean of 2-3 Trans- and Spont-pyMT mice from 6 independent experiments. A slight difference was found in the proportion of CD69⁺ CD8⁺ T cells after DMXAA + anti-TGFβ treatment compared to DMXAA in the TDLN of Spont-PyMT mice (p=0.03, student *t* test).

Overall, we conclude that an anti-TGF β treatment unlocks the production of type I IFN by tumor-infiltrating macrophages when DMXAA is injected in Spont-PyMT mice. This allows for regression of the tumor by a mechanism independent of CD8 T cells. Most probably, the release in the tumor microenvironment of type I IFN, together with TNF α , has a direct cytotoxic effect on the tumor vasculature and the tumor cells (*Spaapen et al., J Immunol 2014*). Tumor infiltrating macrophages play an important role in the production of these cytokines (*Fridlender et al., Br J Cancer 2013; Weiss et al., OncoImmunol 2017*). By contrast, the larger regression induced by DMXAA in Trans-PyMT was dependent on both CD8 T cells and macrophages (*Weiss et al., OncoImmunol 2017*). It therefore appears that in the GEM model, an anti-TGF β treatment allowed for the stimulation of a conspicuous innate anti-tumor response, which however was insufficient to re-invigorate the adaptive anti-tumor response. In line with this is the observation that the DMXAA-induced regression is more transient in anti-TGF β -treated Spont-PyMT, than in the transplanted model (relapse after 10 days *versus* 15 days, respectively).

Nevertheless, Spont-PyMT tumors are not “cold tumors”, since they are infiltrated by some CD8 T cells (**Figure 1d of the manuscript and Figure 4 above**). Recently, the combination of anti-TGF β with anti-PD1 blocking Abs was shown to change drastically the activation of tumor-associated fibroblasts in a way that facilitates infiltration of CD8 T cells in tumor islets (*Tauriello et al., Nature 2018; Mariathasan et al., Nature 2018*). Within the time frame available to answer all comments, we could treat only one extra Spont-PyMT mouse with the triple combination of anti-PD1, anti-TGF β and DMXAA, but measurements of the 8 tumors in this animal did not show a further improvement of the regression observed with a combination of DMXAA and anti-TGF β .

In the revised version of the manuscript, **we have revised Figure 5e and the results on pages 11, 12** to illustrate that the tumor regression observed with the combined DMXAA + anti-TGF β treatment of Spont-PyMT was completely dependent on the IFN pathway, but was independent of CD8 T cells, and **we have further discussed the potential cellular mechanisms of this combined therapy in the discussion (page 17)**.

4. p4: There is no need to editorialize that “Ming et al nicely described...”. It is preferable just to state the data

We have modified the text accordingly.

Reviewer #2: Breast Cancer and microenvironment

(Remarks to the Author):

1. [...] it is important for the authors to indicate total tumor burden and explicitly state what %change means in Figure 1 growth response (e.g. index lesion, per palpable tumor, total tumor volume). Indeed, it is only at the end of the last figure legend that the authors state that “Data are derived from 3-5 mice from 3 independent experiments.” Indicating that tumors growth is not independent measurements (i.e. within same mouse). This is a major limitation of the study, especially in light of the introduction in which the authors lead the reader to consider the evolving tumor microenvironment, which is an important determinant, but is not controlled for in this study in which a mouse with a single tumor is compared to mice with multiple tumors and metastatic disease. This aspect of the paper should be reviewed by a statistician.

We agree with the reviewer that a comparison between transplanted and spontaneous mammary tumor models is tricky because these models differ in many respects, including the number of tumors per animal (one, in one case, and up to ten in the other). The variables that differ in the two models cannot easily be examined separately, and in addition, in the GEM model, all the tumors of a given mouse do not grow at the same rate. In the GEM model, in order to get the most solid data, we considered best to deal with the global tumor burden for each mouse. However, such a representation erases the heterogeneity in the growth of individual tumors in the same mouse, which should also be taken into account.

We agree that our initial representation for the tumor growth evolution was not optimal, and we have worked at improving it. We have revised the figures of the manuscript, as suggested by the reviewer. More precisely, for each treated mouse, we have calculated the total tumor burden (**as shown in Figure 5a below**), corresponding to the sum of the volume of all palpable tumors (8-10 for each mouse). In this Figure, each curve corresponds to a mouse. For the sake of clarity, we have made a second presentation of the data (**Figure 5b below**), showing the relative average changes in tumor lesions from baseline, calculated for each mouse by dividing each lesion size at the indicated time point, by the size of the corresponding lesion at day 0 (DMXAA injection), then averaging the fold change in global lesion size for each mouse at different time points. Again, each curve corresponds to a mouse, **in Figure 5b as in Figure 5a**. This second representation allows to take into account the evolution of all the lesions and not mostly of the largest ones. Finally, in order to compare the responses observed in Spont-PyMT mice to the transplanted PyMT model, we have plotted at day 5 after treatment (**Figure 5c below**), the percentage of mice that showed an overall progression (average changes in tumor lesion >20% from baseline), stabilization (average changes -

30% < x < 20%) or regression (average changes > -30%). These representations are now provided in the revised Figures 1, 5 and S3 of the manuscript.

Figure 5: Representation of the tumor growth evolution after DMXAA treatment. (a) Evolution of the total tumor burden in individual Spont-PyMT mouse treated with one i.p. injection of DMSO (ctrl, n=28 mice) or DMXAA (n=30) from 10 independent experiments. (b) Average changes in the size of tumors compared to baseline measured in each treated mouse. (c) The percentage of mice showing overall regression, progression or stabilization according to the average changes in the size of their tumors, is represented.

2. Throughout the manuscript, the experimental approaches are not self-evident as the figures move from *in vivo* to *in vitro*, injection to stimulation, protein to mRNA without explanation leaving the reader to puzzle out by reference to methods and figure legends what was done, which is still often unclear. In some cases, the methods are in the figure legends (e.g. Fig 3C legend), rather than organized in the methods. In the FACS analysis, immune infiltrate is reported as % viable cells, which includes tumor cells. Thus changes do not reflect the immune infiltrate per se but rather are confounded by the growth rate of the tumor, clearly different in the spontaneous and transplanted tumors. Infiltrate would be more informative if normalized to CD45⁺ cells. In this regard, immune cell phenotypes are presented after the fact so to speak, for example in Figure 3, the authors conclude that DMXAA does not stimulate IRF3 phosphorylation in tumor cells and immune infiltrates in Spont-PyMT tumors but then go on to state the monocytes population are distinct, which is the fundamental basis for the difference.

We thank the reviewer for these remarks, which have helped us to improve the clarity of the manuscript, and the ease to read it. **We have entirely revised the results and reorganized the figures** in order to clearly separate the results based on the *in vivo* treatments from the experiments conducted with restimulation *in vitro* to address the mechanisms underlying this type I IFN defect in Spont-PyMT mice. **We have also rewritten the legends** to improve consistency in the description of the experimental procedures or description of the figures. We hope that the revised version is much clearer now. Following the suggestion of the reviewer, **we have also modified Figure 1 (in c and d)** to provide the percentage of TAM, neutrophils, monocytes or CD8 T cells among CD45⁺ cells, as well as the percentage of CD45⁺ cells in the treated tumors. **The results have been modified accordingly (page 6).**

3. Given that TGF β is well-established mediator of monocyte differentiation and function, the next experiment to show that its inhibition shifts macrophage polarization is predictable, but does not support the Title "TGF blocks STING-induced IFN / r..." as what TGF β does is to shift the myeloid population. Also the legend for figure 4 "The accumulation of TGF in Spont-PyMT but not in Trans-PyMT tumors impaired the activation of IRF3 and the production of IFN after DMXAA activation" is not supported by the data presented therein.

We agree with the reviewer that TGF β is one of the many factors able to influence monocyte differentiation. However, several key points were not predictable 1) That TGF β is much more abundant in the GEM than in the transplanted model 2) That a major consequence of this abundance is the effect exerted on the macrophage ability to produce IFN, these cells being key IFN producers in the tumor. We agree that the initial title had to be improved, as it was too elliptical. **The new title is closer to the experimental evidence: "TGF β blocks IFN α / β release and tumor rejection in**

spontaneous mammary tumors". In addition, in the revised version of the manuscript, **we bring mechanistic data (Figure S5 and pages 10, 11, 15 and 16)** showing how TGF β blocks type I IFN response at the molecular level in macrophages (**please see above our reply to the question 2 raised by the Reviewer 1**). **We have also modified the legend of the Figure 4** that was indeed showing pSmad2/3 staining in the Spont-PyMT tumors and not the defect in pIRF3.

4. In Figure 5, the authors conclude that "the neutralization of TGF in vivo allowed the DMXAA-induced phosphorylation of IRF3" but given the 5 day pre-treatment and the resulting shift in polarization, this is misdirection that is then the basis for demonstration of the expected effect.

The fact that an anti-TGF β could influence myeloid cell differentiation could indeed be expected. However, its consequence, i.e., the observation that "*the neutralization of TGF β in vivo allowed the DMXAA-induced phosphorylation of IRF3*" was by no means expected. We always had in mind that the anti-TGF β pretreatment on pIRF3 could be related to the activation status of the infiltrating macrophages, but we ignored how important was their role. Indeed, it had never been reported that activated MHC II⁺ macrophages after anti-TGF β treatment would be the main cells in which IRF3 was phosphorylated after STING or TLR4 stimulation. To follow the reviewer's suggestion and to be as clear as possible, **we have modified the sentence to "the pretreatment of mice with anti-TGF β in vivo allowed DMXAA to induce the phosphorylation of IRF3 in a fraction of cells" in the revised version of the manuscript (page 10).**

Our interpretation is that TGF β signaling in macrophages induces many molecular changes in these cells (or "*polarization*"), including the nuclear to cytoplasm export of HDAC4 as mentioned above. These changes negatively regulate the capacity of the macrophages to phosphorylate IRF3 and produce type I IFN. **In the revised manuscript, we have discussed this point (pages 15 and 16)** in order to clarify how the anti-TGF β treatment may work.

5. Finally, the penultimate experiment of DMXAA + anti-TGF β lacks controls (i.e. TGF β neutralizing antibody alone, and irrelevant control antibody) as well as test against the transplanted tumor.

Following the reviewer comments, we have treated additional Spont-PyMT mice and transplanted PyMT mice with 1) anti-TGF β alone, 2) DMXAA + isotype control antibody, in parallel to 3) DMXAA +

anti-TGF β or 4) isotype control antibody. Treatment of Spont-PyMT mice with anti-TGF β alone slowed down tumor growth but did not induce tumor regression as compared to the combined treatment DMXAA + anti-TGF β (**Figure 6 below right panel and Figure 5e of the revised manuscript**). Moreover, in the transplanted tumor model, the addition of anti-TGF β did not improve the efficacy of DMXAA treatment (**Figure 6 below, left panel**).

Figure 6: An anti-TGF β treatment has some anti-tumoral effect in Spont-PyMT mice but not in transplanted PyMT mice. Fold change in target lesions from baseline in Spont-PyMT mice (left) or transplanted PyMT mice (right) after treatment with anti-TGF β or isotype control combined with one i.p. injection of DMSO (ctrl) or DMXAA. Data represent the mean per treatment group. Data are from 3-4 mice/group, from 3 independent experiments.

- Also the experiment is only carried out 5 days post treatment, which is insufficient to assess immunological mediated regression. Note too that the same degree of regression is seen in DMXAA treated alone waterfall plot, but fewer tumors did so.

This point has been answered above, in our answer to Reviewer 1, question 3.

- Finally, the authors do not actually test their fundamental conclusion that STING is the essential mediator of tumor regression, nor that the difference between spontaneous and transplanted tumors is dependent on it.

There is already a strong evidence in the literature showing that DMXAA is a ligand of STING, and that it activates a signaling cascade: DMXAA \rightarrow STING \rightarrow IRF3 phosphorylation \rightarrow type I IFN secretion (plus TNF α and cytokines). To provide further evidence that DMXAA-induced tumor regression is STING-dependent and is blunted in the Spont-PyMT model, we have treated Spont-PyMT and transplanted PyMT mice with another STING agonist, ML RR-S2 CDA, injected directly into the tumors, three times as described previously (*Corrales et al., Cell report 2015*). This STING-induced

tumor regression was observed again in the transplanted PyMT model but not in the Spont-PyMT model. **These data are presented in Figure S3 and page 9 of the revised manuscript.** The tumor regression induced by DMXAA or by the other STING agonist relies on the capacity of tumor bearing mice to phosphorylate IRF3 and to produce type I IFN in the tumor microenvironment.

Finally, we wondered whether the reviewer had noticed that the TLR4-induced type I IFN was also altered in this Spont-PyMT model (**Figure 2c**). We would agree with the idea that in the GEM model, the absence of regression is more generally dependent on IRF3 than necessarily on STING. **Such a notion is reflected in the new title of the paper: "TGF β blocks IFN α/β release and tumor rejection in spontaneous mammary tumors".**

Reviewers' comments:

Reviewer #1 (Remarks to the Author):

The revised manuscript by Guerin and colleagues is substantially improved as a result of the revision. The new data are persuasive and the conclusions are now much firmer. I only have one remaining suggestion, which is to describe the anti-tumor data with i.t. STING agonist injection within the opening paragraphs of the results. The fact that a similar effect is seen will be more persuasive for the readers who have been utilizing i.t. administration in general.

Reviewer #3 (Remarks to the Author):

This study by Gurin et al. compare MMTV-PyMT spontaneous vs transplant models in their response to DMXAA-induced tumor regression. They found MMTV-PyMT spontaneous tumors are resistant to DMXAA treatment, due to inactivation of STING mediated pathways. The authors further showed TGFb blockade is a key to DMXAA-induced tumor regression by polarizing TAM into M1 phenotype.

1. Readout of tumor growth, the MMTV-PyMT spontaneous tumors are multifocal involving the entire mammary gland. The authors need to show in Figure 1 how the tumor growth is evaluated, e.g. number & size of foci, total tumor weight etc. % changes do not provide detailed scientific evaluation. This problem also applies to the last figure, which lacks statistical analysis.

2. The authors propose MMTV-PyMT spontaneous tumors are resistant to DMXAA treatment, due to inactivation of STING mediated pathways (as the authors said: In sharp contrast with the tumor regression systematically induced by DMXAA in transplanted PyMT mice, we report here that Spont-PyMT tumors rarely regress after STING activation). TGFb blockade is critical to DMXAA-induced tumor regression

If the authors propose the M1 macrophage polarization is critical in INFa/b production, then these cells need to be depleted to establish the cause-effect relationship.

If mechanisms of TGFb blockade in potentiating DMXAA efficacy in tumor regression are mediated through the Sting-IRF3-INFa/b pathway, then the authors may consider treating the Spont-PyMT with a STING antagonist, which should reverse the effect of TGFb blockade + DMXAA treatment. If both mechanisms are acritical, then the authors need to show data to support. In short, the author did not provide strong evidence rather only correlative studies. This flaw undermines the author's conclusion in their effort comparing the spontaneous vs transplanted tumors.

3. In Figure 2, the relative mRNA expression of critical cytokines and chemokines are weak evidence. Protein expression and real data (rather than "relative") are preferred

4. Spatial location of the immune cells in the tumor tissues should be evaluated, which is an important factor in effective immune response.

Minor point: in Figure 4c, the red IF was labeled as pSmad 2/3 in photo, but pIRF3 in legend, this mistake actually affects the conclusion; Figure 5a, no control for comparison

To reviewer #1: We are very pleased to read that the reviewer considers our manuscript as substantially improved as a result of the revision. This improvement is due in part to the quality of his/her initial comments on this work. **We have of course taken into account the remark on the opening paragraph of the results (see page 6 and Figure 1d).**

To reviewer #3:

1) For the *readout of tumor growth*, reviewer #3 did not really take into account the queries of reviewer #2. In the initial version of the manuscript, tumor growth was expressed tumor by tumor. Reviewer #2 asked that this growth should be presented with one curve per mouse, and we had followed his recommendation in our first revision. Such a representation is tricky to obtain, since in the same mouse it is possible to see some tumors growing, whereas other were decreasing. We spent a long time choosing the representation finally adopted, in order to take into account both the requirement of reviewer #2 and the variability within a mouse. Reviewer #3 does not comment our answer to reviewer #2. Instead, he asks us to represent the evolution of the size of each tumor, which is precisely the opposite of what was requested by reviewer #2. In addition, he requires us to provide the number of foci, which is of weak significance, since it changes very little with the evolution of the tumors (as indicated in the manuscript, mice are treated when they have already developed about 6-8 palpable tumors). The last reviewer's request for representing the evolution of the tumor weight is strange, since the total tumor weight can only be an end point, obtained when mice are killed. It cannot be represented as a function of time. The only thing that can be represented as a function of time is an evolution of the global tumor volume, which is precisely what we did...

However, we have been willing to further clarify the way tumor growth was evaluated, as requested by reviewer 3. To this end, **we have revised Fig. 1, and provide first an illustration of the tumor size evolution of individual tumors in one mouse (new Fig. 1a), before continuing as before.**

Regarding the statistical analysis for the last figure (Figure 6), **below are the graphs showing (a) the cumulative distribution for each treatment group, (b) the p values after running a Kolmogorov-Smirnov test, and (c) the significant p values.** We hope that this will make things clearer. **The p values are provided in supplementary table 2 of the revised manuscript.**

p-values from Kolmogorov-Smirnov test

c

Significant p-values ($p < 0.05$) from Kolmogorov-Smirnov test

2) The reviewer says *"If the authors propose the M1 macrophage polarization is critical in INF α /b production, then these cells need to be depleted to establish the cause-effect relationship"*. This comment is surprising. Indeed, Fig. 6C was precisely showing the result of requested experiment, as it demonstrates that DMXAA-induced INF α production is no longer observed when anti-TGF β treated tumors are depleted in macrophages. In addition, the analysis of macrophage depletion experiments, in vivo, in the transplanted tumor model, has been published in our previous paper (Weiss et al., 2017), which is abundantly quoted in the present manuscript.

Nevertheless, to strengthen this conclusion, **we have revised the manuscript to provide new experiments with in vivo depletion of myeloid cells. This is now illustrated in Fig. S4. We have modified the results section accordingly (page 10).**

Reviewer #3 asks us to perform new experiments with a putative STING antagonist. This is a *new* question, which had been raised by none of the original referees. Moreover, an experimental answer to this new query is unlikely to yield a meaningful outcome. We have provided strong experimental evidence that the well characterized STING ligand (Conlon et al., 2013; Prantner et al., 2012), DMXAA, triggers all the events expected from a STING ligand : an increase in pTBK1 and pIRF3, and the production of type I IFN. Moreover, we have demonstrated that an anti-TGF β treatment alone does not induce tumor regression as shown in Fig. 6d. **We also provide new data in the revised version demonstrating that TGF β does not induce type I IFN on its own (Fig S4b).**

In these conditions, it is difficult to imagine what could be gained from testing the simultaneous effect of an agonist plus an antagonist of STING. Answering this *new* request would represent a lot of work (since we have no bona fide STING antagonist available). This will cause a further delay in our publication, which has already suffered excessive delays. All this for no meaningful outcome.

The reviewer writes *"If both mechanisms are acritical, then the authors need to show data to support. In short, the author did not provide strong evidence rather only correlative studies"*. This sentence is rather obscure to us. However, it seems that the reviewer raises a new objection: all our findings only correspond to correlative studies. A demanding and careful reader of this work, such as reviewer #1, would not make (and has not made) such a claim.

3) The reviewer says that *the relative mRNA expression of critical cytokines and chemokines are weak evidence and Protein expression and real data (rather than "relative") are preferred*". This remark may seem unfair since the quantity of INF α protein, in in vitro experiments, is depicted in several figures, in addition to the detection of pIRF3, as well as pTBK1, and pSMAD2/3 for key pathways. For direct ex vivo measurements, mRNA expression is obviously the method of choice. Moreover, if, in Spont-PyMT tumors, there is no increase in INF α mRNA, and no tumor regression, it would be strange to expect an increase in INF α protein... Despite this, **we show below the detection of chemokines at the protein level, measured in tumors from Spont- and Trans-PyMT tumors of control and DMXAA treated mice, leading to similar conclusions as in Figure 3 and Figure S1.**

These new data show that, even if there is no DMXAA-induced IFN α and tumor regression in Spont-PyMT mice, DMXAA has a partial action on TNF α and some chemokines, which turns out to be functionally inefficient. We had already reached this conclusion with mRNA. The protein data provide a mere confirmation of our former conclusion.

Detection of cytokines /chemokines in the tumors of Spont-PyMT and Trans-PyMT at the protein level. Mice were treated with DMXAA (1x i.p., 23mg/kg) or DMSO 50% as control. Mice were sacrificed after 24h and the cytokines/chemokines were measured by multiplex in the supernatant of tumor slices after overnight culture. Data are from Spont-PyMT mice: $n=3$ DMSO and 3 DMXAA; Trans-PyMT mice: $n=3$ DMSO and 3 DMXAA, collected in 2 independent experiments. Results are expressed as mean \pm s.e.m. Student t -Test.

4) The reviewer says that "*Spatial location of the immune cells in the tumor tissues should be evaluated, which is an important factor in effective immune response.*"

We fully agree with the reviewer that the question of immune cell localization is too often overlooked, albeit not by us. Indeed, such a question has been examined in all our previous and future papers (Thoreau et al., 2015; Weiss et al., 2017; Guerin in preparation). This point will be further deepened in a review on murine tumor models that we will publish soon, in which we discuss extensively the question of the presence of immune cells in the stroma and in tumor islets. However, examining it in the present paper would be premature. The present paper shows that an excess of TGF β in spontaneous tumors explains part of the resistance of these tumors to anti-tumoral treatments. This is already a major success, which will deserve to be completed for overcoming other sources of resistance present in spontaneous tumors. When we will be able to combine anti- TGF β and another treatment resulting in a further enhanced anti-tumor efficacy, then it will be time to examine in detail where immune cells are found in the tumor, before and during treatment-induced strong regressions. We have not reached this point yet.

The minor points for the legend of Figure 4c has been corrected (page 30) and the control for Figure 5a is illustrated in Figure 4a.

References:

Conlon, J., Burdette, D.L., Sharma, S., Bhat, N., Thompson, M., Jiang, Z., Rathinam, V.A.K., Monks, B., Jin, T., Xiao, T.S., et al. (2013). Mouse, but not human STING, binds and signals in response to the vascular disrupting agent 5,6-dimethylxanthenone-4-acetic acid. *J. Immunol. Baltim. Md 1950* 190, 5216–5225.

Prantner, D., Perkins, D.J., Lai, W., Williams, M.S., Sharma, S., Fitzgerald, K.A., and Vogel, S.N. (2012). 5,6-Dimethylxanthenone-4-acetic acid (DMXAA) activates stimulator of interferon gene (STING)-dependent innate immune pathways and is regulated by mitochondrial membrane potential. *J. Biol. Chem.* 287, 39776–39788.